



# Amplified bottom water acidification rates on the Bering Sea shelf from 1970-2022

Darren J. Pilcher[1,2], Jessica N. Cross[2,3], Natalie Monacci[4], Linquan Mu[1], Kelly A. Kearney[1,5], Albert J. Hermann[1,2], Wei Cheng[1,2]

[1] Cooperative Institute for Climate, Ocean, and Ecosystem Studies, University of Washington, Seattle, WA, USA

[2] NOAA Pacific Marine Environmental Laboratory, Seattle, WA, USA

[3] Pacific Northwest National Laboratory, Sequim, WA, USA

[4] University of Alaska Fairbanks, Fairbanks, AK, USA

[5] NOAA Alaska Fisheries Science Center, Seattle, WA, USA

*Correspondence to:* Darren J. Pilcher (darren.pilcher@noaa.gov)

**Abstract.** The Bering Sea shelf supports a highly productive marine ecosystem that is vulnerable to ocean acidification (OA) due to the cold, carbon rich waters. Previous observational evidence

suggests that bottom waters on the shelf are already seasonally undersaturated with respect to aragonite (i.e. $\Omega_{arag}$ < 1), and that OA will continue to increase the spatial extent, duration, and intensity of these conditions. Here, we use a regional ocean biogeochemical model to simulate changes in ocean carbon chemistry for the Bering Sea shelf from 1970-2022. Over this timeframe, surface $\Omega_{arag}$ decreases by -0.043 decade$^{-1}$ and surface pH by -0.014 decade$^{-1}$,

comparable to observed global rates of OA. However, bottom water pH decreases at twice the rate of surface pH, while bottom $[H^+]$ decreases at nearly three times the rate of surface $[H^+]$. This amplified bottom water acidification emerges over the past 25 years and is likely driven by a combination of anthropogenic carbon accumulation and a trend of increasing primary productivity and increasing subsurface respiration and remineralization. Due to this enhanced

bottom water acidification, the spatial extent of bottom waters with $\Omega_{arag}$ < 1 has greatly expanded over the past two decades, along with pH conditions harmful to red king crab. Interannual variability in surface and bottom $\Omega_{arag}$, pH, and $[H^+]$ has also increased over the past two decades, resulting in part from the increased physical climate variability. We also find that the Bering Sea shelf is a net annual carbon sink of 1.1-7.9 TgC/year, with the range resulting

from the difference in the two different atmospheric forcing reanalysis products used. Seasonally, the shelf is a significant carbon sink from April-October but a somewhat weaker carbon source from November-March.



## 1   Introduction

The global ocean presently absorbs 25-31% of annual $CO_2$ emissions, making it a critical
carbon sink that mitigates anthropogenic warming (Gruber et al., 2019; Friedlingstein et al.,
2020; McKinley et al., 2020).  The uptake of this anthropogenic carbon has driven a shift in the
marine carbonate system towards a state of lower pH and carbonate saturation, a process referred
to as Ocean Acidification (OA; Feely et al., 2004).  High latitude regions are particularly
vulnerable to OA due to the poorly buffered, cold temperature waters generating naturally low
carbonate saturation states (Fabry et al., 2009).  Experimental studies have determined a number
of negative effects to marine organisms due to OA (Doney et al., 2020), particularly for
organisms that form calcium carbonate shells as these shells become harder to build and maintain
as carbonate saturation states ($\Omega$) approach and drop below 1.  Pteropod shell dissolution has
already been observed in several high-latitude environments (Bednarsek et al., 2012; Niemi et
al., 2021), and OA is expected to shift these conditions equatorward over time.

Although OA is driven by the increase in atmospheric $CO_2$ and subsequent increase in
ocean carbon uptake, there are a number of physical and biogeochemical processes that can
modify the rate of OA expected from the increase in atmospheric $CO_2$ (Hauri et al., 2021).  For
example, the accumulation of respired carbon at depth reduces the buffer capacity of subsurface
water, leading to amplified subsurface acidification rates compared to surface waters throughout
large regions of the global oceans (Fassbender et al., 2023).  Coastal shelf systems can
experience local rates of acidification much faster than the global oceans due to upwelling (Feely
et al., 2008), biological respiration (Feely et al., 2010), eutrophication (Laurent et al., 2017), and
changes in circulation (Siedlecki et al., 2021).  In the Arctic, changes in sea ice formation (Zhang
et al., 2020) and biological productivity and remineralization (Qi et al., 2022) can generate
acidification rates 2-3 times greater than the rate for the open oceans.

The Bering Sea is composed of a relatively large (> 500km wide and > 100km long),
shallow eastern coastal shelf along with, a narrow western shelf, and a deep interior basin.  The
shelf itself is composed of three distinct biophysical domains (inner, middle, and outer) often
delineated by the 50m, 100m, and 200m isobaths. General circulation on the shelf tends to follow
these isobaths in a north-northwest direction, eventually feeding into the western intensified
Anadyr Current, which then flows through Bering Strait, thereby providing a key conduit
between the Bering Sea and Arctic (Kinder et al., 1986; Stabeno et al., 2016).  The Bering Sea



shelf ecosystem is strongly tied to the atmospheric and oceanic physical forcing, with the

seasonal formation and retreat of sea ice playing a fundamental role through the development of the bottom water cold pool and by setting the timing and magnitude of the spring phytoplankton bloom (Brown and Arrigo 2013; Sigler et al., 2014). While the formation of sea ice occurs annually, the areal extent and timing of ice formation and retreat can vary substantially. This variability during the past 10-20 years has consisted of multi-year periods of persistent warm,

low sea ice extent (e.g. 2001-2005 and 2014-2018) or cold, high sea ice extent conditions (e.g. 2007-2013; Stabeno et al., 2012). The recent warm years have generated record breaking low sea ice extent and high temperatures in the northern Bering Sea, with substantial negative impacts to the marine ecosystem (Stabeno and Bell, 2019; Siddon et al., 2020).

        On annual timescales, the Bering Sea shelf is generally considered a net carbon sink,

driven by substantial spring-summer primary productivity generating low surface ocean $p\mathrm{CO}_2$ values and a net influx of carbon from the atmosphere (Bates et al., 2011; Cross et al., 2014; Pilcher et al., 2019). A portion of the carbon fixed by this mixed layer productivity sinks to bottom waters where it is respired into inorganic carbon and can be re-emitted back to the atmosphere in fall-winter due to strong atmospheric wind speeds and vertical mixing (Cross et

al., 2014; Pilcher et al., 2019). Sea ice further impacts the seasonal carbon cycle by acting as a physical barrier inhibiting air-sea gas exchange. Furthermore, sea ice formation can pump DIC and total alkalinity to the bottom along with salinity via brine rejection, while sea ice melt dilutes both variables in surface waters (Mortenson et al., 2020).

        Previous observational and modeling studies have found that seasonal periods of

undersaturation of aragonite ($\Omega_{\mathrm{arag}} < 1$) are already occurring within subsurface waters and near regions of significant riverine freshwater runoff (Mathis et al., 2011; Cross et al., 2013; Pilcher et al., 2019). Subsurface $\Omega_{\mathrm{arag}} < 1$ waters occur in summer and early fall, driven by bacterial respiration associated with remineralization of sinking organic matter, particularly in regions of high primary productivity in the middle and outer shelf domains (Mathis et al., 2011). Surface

waters generally maintain much higher values of $\Omega_{\mathrm{arag}}$ and pH due to this significant primary productivity, except near freshwater runoff, particularly the mouths of the Yukon and Kuskokwim rivers, where $\Omega_{\mathrm{arag}} < 1$ and relatively low pH values are driven by relatively high DIC:TA ratios due to terrestrial carbon exports (Mathis et al., 2011; Pilcher et al., 2019). Furthermore, model simulations suggest that winter surface $\Omega_{\mathrm{arag}}$ values are relatively low and



close to 1, particularly in ice covered regions where entrained subsurface carbon cannot re-
equilibrate with the atmosphere (Pilcher et al., 2019). Winter observational data is extremely
sparse due to challenging weather and sea ice conditions; however, limited late-fall data suggest
supersaturated $p$CO$_2$ conditions (Cross et al., 2014; Cross et al., 2016). Model simulations
project that seasonal periods of surface $\Omega_{arag}$ undersaturation may grow to encompass up to 5

months of the year following the RCP 8.5 emissions scenario and 2-3 months following the RCP
4.5 scenario (Pilcher et al., 2022).

The Bering Sea sustains a substantial U.S. fishery, representing 40% of U.S. total fish
catch by weight and $3 billion in annual value (Wiese et al., 2012). These fisheries also provide
commercial, subsistence, and cultural benefits to many Alaskan communities, putting them at

risk from ocean acidification (Mathis et al., 2015). In the Bering Sea, red and tanner crab have
emerged as species particularly vulnerable to the direct effects of OA. The growth rates and
survival of larval and juvenile crab for both species are decreased at lower pH values (Long et
al., 2013a,b; Long et al., 2016). Incorporating these results into bioeconomic models suggests
that the red king crab fishery could substantially decline if OA is not accounted for in the

fisheries management process (Seung et al., 2015; Punt et al., 2016). Recent closures of the
snow crab fishery and the Bristol Bay red king crab fishery have had devastating impacts to the
Bering Sea commercial fishing community and has led to some discussion concerning the
potential role of OA. However, recent laboratory studies have found that snow crab appear
resilient to OA (Algayer et al., 2023), and that the snow crab fishery collapse may be due to a

mass mortality event resulting from the 2018-2019 heatwave (Szuwalski et al., 2023). In
comparison to the collapse in snow crab populations, the Bristol Bay red king crab fishery has
been in a steady decline since 2014 (Fedewa et al., 2020). Although model results suggest that
bottom waters in parts of Bristol Bay have pH values harmful to larval and juvenille red king
crab, these crab populations tend to inhabit nearshore regions that are relatively well buffered

with much higher pH values (Pilcher et al., 2022). Thus, the potential role of OA in impacting
Bristol Bay red king crab populations is currently unclear.

Recent work utilized a regional ocean biogeochemical model and a dynamical
downscaling technique to generate long-term projections of OA for the Bering Sea shelf using
multiple Earth System Models (ESMs) and emissions scenarios (Pilcher et al., 2022). Here, we

greatly expand the temporal coverage of our previous model hindcast (e.g. Pilcher et al., 2019) to



simulate 53 years (1970-2022) of the Bering Sea marine carbon cycle. We use this model output to quantify spatial-temporal trends in Bering Sea shelf marine carbonate variables over the entire hindcast and the underlying mechanisms generating heterogeneity in these trends. We conclude by illustrating how this model output is being incorporated into the fisheries management process and the next steps to continue refining these model-based OA products.

## 2   Methods

### 2.1 Base model description

The regional Bering10K model is an implementation of the Regional Ocean Modeling System (ROMS; Shchepetkin and McWilliams, 2005; Haidvogel et al., 2008), with 10 km horizontal resolution and 30 vertical layers. The Bering10K model simulates sea ice formation and melt, along with tidal mixing. A thorough description of the physical model can be found in Hermann et al., (2016) and Kearney et al., (2020). This physical model is coupled to a lower trophic NPZD model, originally developed as part of the Bering Sea Ecosystem Study (BESTNPZ; Gibson and Spitz 2011), and recently updated by Kearney et al., (2020). Briefly, the BESTNPZ model simulates two phytoplankton groups (small and large), five zooplankton groups (microzooplankton, small copepods, large copepods, euphausiids, and jellyfish), three nutrient groups (nitrate, ammonium, iron), and two detrital groups (slow and fast sinking). BESTNPZ also contains an ice biology sub-model which simulates ice algae, nitrate, and ammonium, along with a benthic sub-model which simulates a benthic infauna group and a detrital group. A thorough description of the BESTNPZ model can be found in Kearney et al., 2020.

Carbonate chemistry is incorporated into the Bering10K BESTNPZ model by simulating dissolved inorganic carbon (DIC) and total alkalinity (TA), which are used to calculate the remainder of the carbonate system following the OCMIP-2 protocols (Orr et al., 1999) and CO2SYS (Lewis and Wallace, 1998). Here we report pH and $[H^+]$ values on the total scale. DIC is generated from planktonic respiration and detrital remineralization, and consumed via planktonic photosynthesis. Additionally, DIC is exchanged with the atmosphere depending on the gradient in the partial pressure of $CO_2$ between the surface ocean and the atmosphere ($DpCO_2$) and the wind speed following Wanninkhof et al., (2014). The atmospheric $CO_2$ concentration is set to the monthly in-situ concentration from Barrow, Alaska (Thoning et al.,



2022). This timeseries started in 1973; for 1970-1972, we take the 1973 Barrow monthly

timeseries and subtract the respective annual growth rate from the Mauna Loa timeseries

(https://gml.noaa.gov/ccgg/trends/). Riverine freshwater runoff flux is prescribed following

freshwater discharge data from Alaska and Russia (Kearney, 2019). This river runoff contains

seasonally varying concentrations of DIC and TA following data collected at Pilot Station at the

mouth of the Yukon River (Striegl et al., 2007; PARTNERS, 2010, Pilcher et al., 2019).

       The atmospheric forcing for air temperature, sea level pressure, longwave and shortwave

radiation, u and v winds, specific humidity, and rainfall are provided by a combination of

reanalysis products. For 1970-1994 we use the Common Ocean Reference Experiment (CORE;

Large and Yeager, 2009) forcing, for 1995-2011 the Climate Forecast System Reanalysis (CFSR;

Saha et al., 2010), and for 2011-2021 the Climate Forecast System Operational Analysis

(CFSv2-OA; Saha et al., 2014). Lateral open boundary conditions at weekly resolution for

temperature, salinity, and oceanic velocities (u and v) are derived from the larger scale Northeast

Pacific model (Danielson et al., 2011) for the CORE forcing timeframe, and the CFSR/CFSv2-

OA for CFSR forcing timeframe. Nitrate boundary conditions are monthly climatologies from a

long-term run of the larger Northeast Pacific (NEP-5) ROMS domain (Danielson et al., 2011).

Oxygen initial conditions and monthly boundary conditions are climatologic means from the

World Ocean Atlas 2018 product (Garcia et al., 2018). Water column iron concentrations are

nudged towards empirical climatological profiles.

       The lateral boundary conditions for DIC and TA are calculated via linear regressions with

salinity, derived from observational data collected primarily from 2008-2010 (Pilcher et al.,

2019). The salinity-DIC regression has changed over time as the oceanic uptake of $CO_2$ has

increased the DIC concentration of waters, with no effect on salinity. Thus, using this same

relationship for the boundary conditions at the start of the hindcast in 1970 would introduce a

high DIC bias. To account for changes in DIC over time, we center the DIC-salinity relationship

on the year 2009 (i.e. midpoint of 2008-2010 sampling timeframe) and subtract (add) DIC for

years before (after) 2009. The DIC value added or subtracted is calculated from the linear trend

in DIC calculated from the historical runs of the Coupled Model Intercomparison Phase 6

(CMIP6) over the 1970-2009 timeframe from the mean of three different Earth System Models.

These three ESMs were selected as they have been coupled previously with the Bering10K



regional model (Cheng et al., 2021; Pilcher et al., 2022). We chose to use this method to gain the higher spatial resolution, particularly in the vertical, provided by the ESM output.

Initial conditions for the start of the hindcast in 1970 for non-carbonate chemistry variables are taken from a 30-year model spin-up using repeating 2001 forcing (Kearney et al., 2020). Initial conditions for TA are calculated using the same salinity regression used for the boundary conditions. Similarly, the DIC initial conditions use the salinity regression, along with subtracting the same long-term trend used for the boundary conditions. The model is then spun-up for an additional three years using repeating 1970 forcing, at which point the model seasonal $CO_2$ cycle was approximately in balance with minimal year-to-year on-shelf variations. The model hindcast is then started and run continuously for 1970-2022.

## 2.2 Model updates

A new addition to the BESTNPZ model presented in previous work is the inclusion of oxygen cycling following Siedlecki et al., (2015) and Bianucci et al., (2011). Oxygen cycling contains phytoplankton growth as a source, and respiration, remineralization, and nitrification as sinks. Oxygen cycling throughout the water column is governed by the following equation:

$$\frac{\partial O_2}{\partial t} = Phy_i * u_i(Light, N) - resp(Phy_i) - resp(Z_i) - remin(D_i) - Nitrification$$
$$+ advection + diffusion$$

(1)

Surface and bottom oxygen concentrations are further modified through the following equations, respectively:

$$\frac{\partial O_2}{\partial t}\bigg|_{surface} = \frac{V_{O2}}{\Delta z} * \left([O_2]_{sat} - [O_2]|_{z=surface}\right)$$

(2)

$$V_{O2} = 0.251u^2 \left(\frac{S_c}{660}\right)^{-0.5}$$

(3)



$$\frac{\partial O_2}{\partial t}\bigg|_{Bottom} = \frac{1}{\Delta z}\left(W_D \frac{dD}{dz}\bigg|_{z=Bottom}\right) - resp(Ben) - excretion(Ben) - remin(DetBen)$$

(4)

where $Phy_i$ is the phytoplankton group, $u_i$ is the growth rate, *Light* and *N* and the light and nutrient limitations respectively, *resp* is respiration, $Zi$ is the zooplankton group, *remin* is bacterial remineralization, and $D_i$ is the detrital group. For the surface equation (2), $\Delta z$ is the vertical thickness of the grid cell, $[O_2]_{sat}$ is calculated following the equation from Garcia and Gordon (1992), $S_c$ is the Schmidt number, and $V_{O2}$ is the gas transfer velocity following Wanninkhof (2014). For the bottom equation (4), $W_D$ is the detrital sinking rate, *Ben* is the benthic infauna group, and *DetBen* is benthic detritus. The above model equations (1-4) utilize constant stoichiometric molar ratios consisting of C:N = 106:16, $O_2$:N = 138:16 for nitrate fluxes, and $O_2$:N = 106:16 for ammonium fluxes. The complete BESTNPZ model equations are found in Kearney et al., (2020).

### 2.3 Observational data for model validation

To assess overall model skill, we compare model hindcast output to several observational datasets. One of the largest available datasets for carbonate chemistry in the Bering Sea was collected and compiled during the 2008-2010 Bering Sea Ecosystem Study (BEST) and Bering Sea Integrated Research Program (BSIERP). This dataset is particularly valuable due to the large number of discrete DIC and TA samples; these are the prognostic model variables used within the model and therefore provide a direct model-data comparison. These data were typically collected in the spring (April/May) and summer (June/July) seasons, along with a fall (September/October) sample period in 2009. The sampling regime covered a large portion of the U.S. southeastern Bering Sea shelf, including three cross-shelf transects (Fig. 1). $p$CO$_2$, pH, and $\Omega_{arag}$ values were calculated from DIC, TA, salinity, and temperature measurements using CO2SYS (Cross et al., 2012; Cross et al., 2013).

The M2 mooring is the longest dataset for surface ocean $p$CO$_2$ in the Bering Sea. While the M2 mooring itself provides a multi-decadal long timeseries of standard oceanographic properties, the moored autonomous surface vehicle (MAPCO2; Sutton et al., 2019) system used to measure $p$CO$_2$ was first deployed in 2013 and has since been re-deployed with the M2 mooring during the ice-free season for every year except 2020. Generally, this timeseries covers



the months of May-September, however in 2021 it was left out much later than usual, providing

the first glimpse of late fall and early winter $p\text{CO}_2$. For further model validation of $p\text{CO}_2$, we

also utilize $p\text{CO}_2$ measurements from an Autonomous Surface Vehicle $\text{CO}_2$ System (ASVCO2)

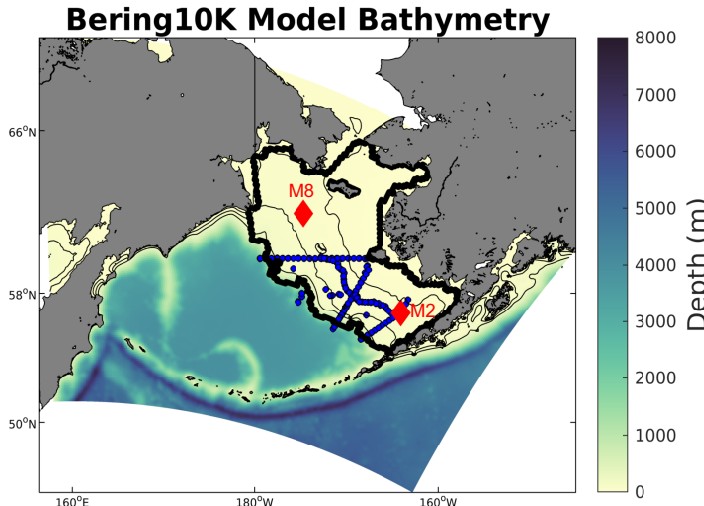

**Figure 1:** Spatial map of the model domain along with the model bathymetry. Also shown are the discrete ship-based sample locations (blue dots) and the two moorings (red diamonds) used for model validation. The black line denotes the spatial region used to encompass the Bering Sea shelf.

onboard the Saildrone uncrewed surface vehicle (USV) (Wang et al., 2022). This dataset

provides a transect of surface ocean $p\text{CO}_2$, generally running from the Aleutian Islands to the

Bering Strait during missions to the Chukchi Sea from 2017-2019. Therefore, each year contains

a northward transect in late spring/early summer, along with a southward transect in late

summer.


### 3 Results

### 3.1 Model Skill Assessment

Model property-property comparisons and associated skill statistics between discrete

samples collected during 2008-2010 and the model hindcast illustrate relatively high correlation

coefficients across the water column for most model prognostic variables (Fig. 2). However, a

slight negative TA bias combined with a slight positive DIC bias work synergistically to generate

a relatively larger negative bias in $\Omega_{\text{arag}}$ and pH. Another notable model-data mismatch is that

subsurface points (depth > 200m) for salinity, $\text{NO}_3$, TA, and DIC are all relatively lower in the

model compared to the observations. These points are all outside of our definition of the Bering



Sea shelf (encompassing depth 0-200m; Fig. 1) and are located on the shelf break, which is

smoothed in the model bathymetry to ensure numerical stability (Kearney et al., 2020).

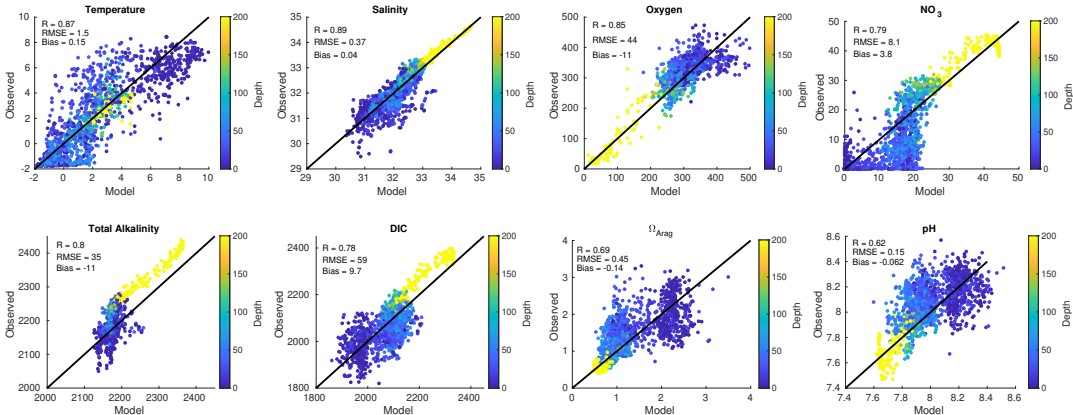

**Figure 2:** Plots of model (x-axis) and observed (y-axis) co-located points for different model variables. Also shown in each plot
are the R, RMSE, and bias skill statistics. Observed data are from the 2008-2010 BEST-BSIERP project, shown as blue dots in
Fig. 1. Note that here the colorbar is constrained to depths between 0-200m because our focus is on the shelf, though deeper, off-
shelf points (denoted by bright yellow dots) are still included.

The model-data comparison illustrated in Fig. 2 is further summarized via a Target

Diagram (Jolliff et al., 2009) in Fig. 3. In a Target diagram, the position in the y-axis denotes

either a positive (Y > 0) or negative (Y < 0) normalized model bias, while the position in the x-

axis signifies whether the model has a larger (X > 0) or smaller (X < 0) root-mean-square-

deviation (RMSD) compared to the observed data. The radial distance from the origin

(normalized RMSD) is then related to the modeling efficiency metric (MEF; Stow et al., 2009),

where model variables that lie within the RMSD < 1 circle have a MEF > 0, signifying that the

model outperforms an estimate based solely on the mean of the observations. Figure 3 illustrates

that all highlighted model variables fall within the RMSD value of 1, with relatively low overall

biases. Most model variables display less variability compared to the observations, except for

$\Omega_{arag}$ which displays more variability.

In addition to the ship-based observational comparison, model output of surface ocean

$p$CO$_2$ is also compared to the M2 mooring timeseries (Fig. 4). The model accurately captures the

timing of the late spring $p$CO$_2$ drawdown along with the subsequent increase in $p$CO$_2$ leading

into summer. Furthermore, the modeled late fall and early winter increase in $p$CO$_2$ is also

apparent in the mooring for the single year that the mooring was left out late into the season.

However, the model generally tends to underestimate the magnitude of the late spring $p$CO$_2$

drawdown, which then subsequently leads to model overestimations of summer $p$CO$_2$. Notable




exceptions are apparent in 2013, 2018, and 2022 when the observed late spring $p$CO$_2$ drawdown
was relatively weaker, and the modeled drawdown is more comparable with observations.

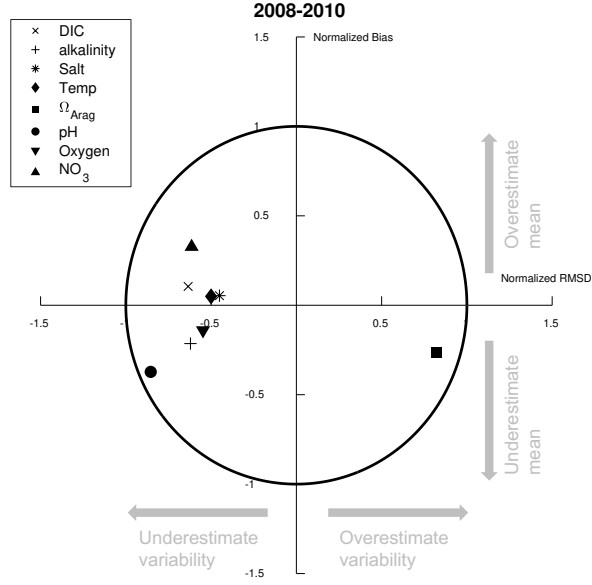

**Figure 3:** Target diagram summarizing the data comparison from Fig. 2. Here, the X-axis is the normalized unbiased RMSD between the model and data, multiplied by the sign of the difference between model and observed standard deviation. The Y-axis
is the normalized mean bias.

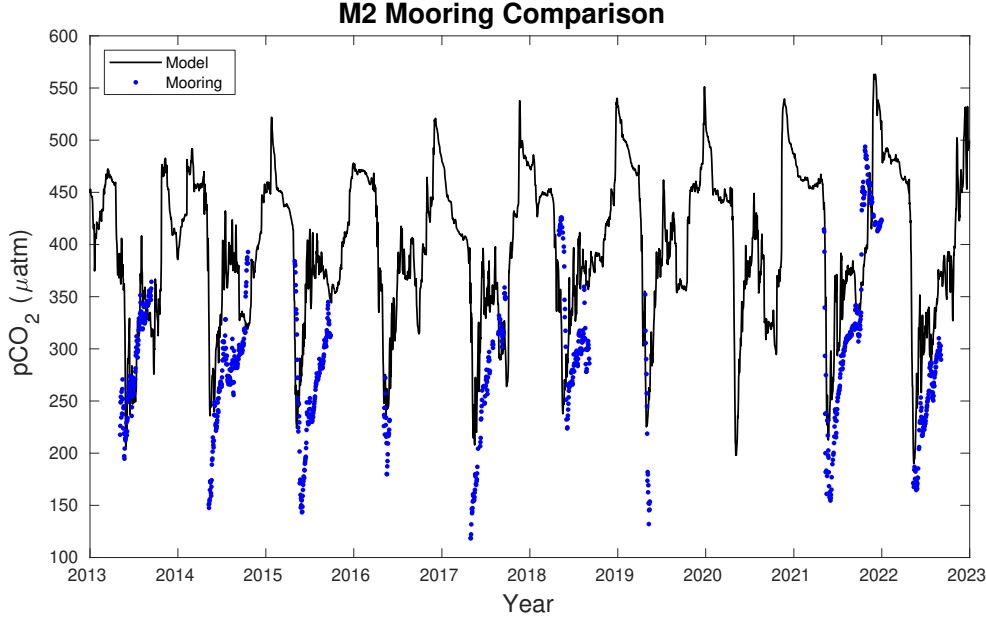

**Figure 4:** M2 mooring $p$CO$_2$ data (blue dots) compared to model daily $p$CO$_2$ values (black line) at the equivalent model grid cell location. The mooring is generally deployed in spring and retrieved in fall, though was out much later in 2021.





Further surface $pCO_2$ comparisons between the model output and in-situ $pCO_2$ from the

autonomous Saildrone platform are shown in Fig. 5. Overall, the model does a reasonably

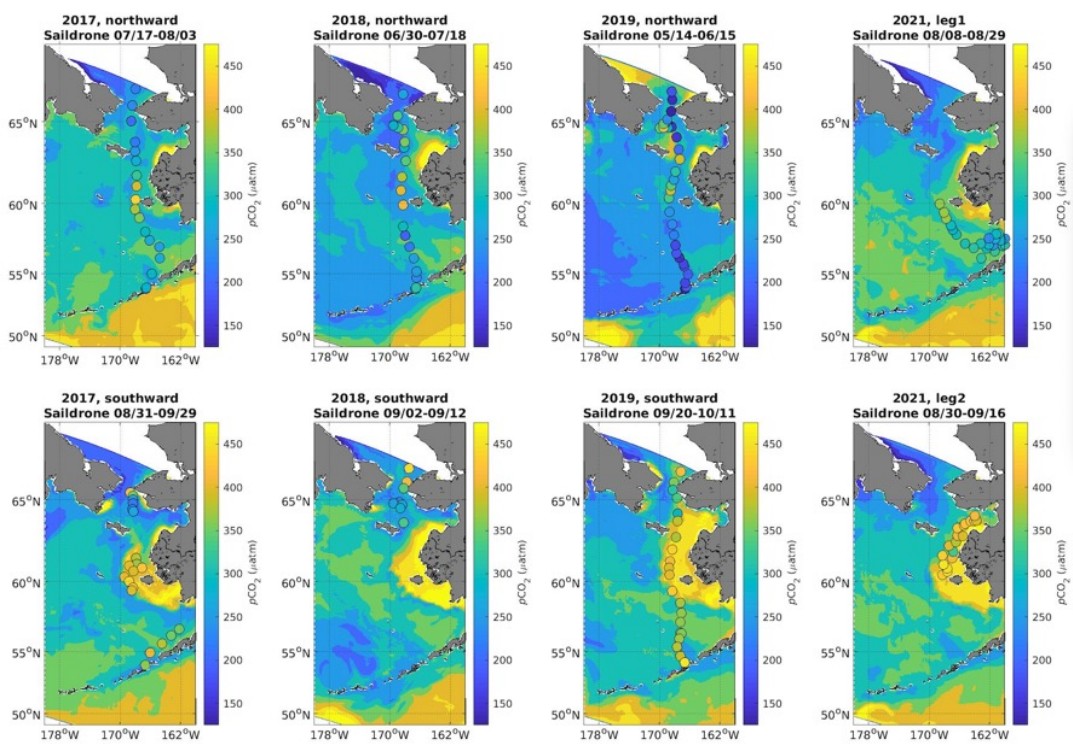

**Figure 5:** Surface pCO₂ values from Saildrone transects (dots) with model surface pCO₂ values averaged over the equivalent timeframe as the background shading.

sufficient job of capturing the dominant spatial pattern in $pCO_2$ illustrated by the Saildrone data,

namely the relatively lower $pCO_2$ values in the southeastern and northern Bering Sea with higher

values in the central inner shelf domain near Nunivak Island. The seasonality between the two

transects also aligns, with relatively lower values during the northward transect and higher values

during the southward transect. However, the model appears to consistently underestimate the

$pCO_2$ drawdown (i.e. model $pCO_2$ biased high compared to Saildrone data) in the southeastern

Bering Sea during the northward transect, similar to the underestimated spring $pCO_2$ drawdown

from the M2 mooring comparison (Fig. 4). However, the southward transects suggest that this

bias is reversed later in the year, where the model is now biased low compared to the Saildrone

data, which is also the opposite bias of what we see during the late summer and early fall in the

M2 mooring comparison. Additionally, the model tends to underestimate $pCO_2$ in the central

inner shelf domain just to the west of Nunivak Island. It appears that the Saildrone data is





consistently capturing a relatively high plume of $p$CO$_2$ in this region. The model also generally simulates these relatively high $p$CO$_2$ waters in that region of the inner shelf domain, but there is a lot of interannual variability and seasonality in this feature.

This analysis suggests that the model is simulating the Bering Sea carbon cycle reasonably well, though there are some noted differences. Namely, the model appears to underestimate variability overall (Fig. 3) and underestimate the magnitude of the seasonal $p$CO$_2$ drawdown according to both the M2 mooring and Saildrone data. This could be due to a somewhat smaller magnitude spring bloom, which is consistent with slight positive model biases in DIC and NO$_3$ from the ship-based observation comparison (Fig. 2). This bias could translate

to model pH and $\Omega_{arag}$ values that are biased low in surface waters but biased high in bottom waters, due to less respiration of sinking organic carbon from a smaller spring bloom. However, we caution that bottom measurements are very limited overall, and were all collected during the anomalously cold-water conditions during 2008-2010. Furthermore, $p$CO$_2$ is a relatively difficult variable for the model to capture because it is a nonlinear, diagnostic variable that is

dependent on temperature, salinity, DIC, and TA. This nonlinearity and the potential for synergistic biases (e.g. positive DIC bias but negative TA bias) can generate very large magnitude deviations. Thus, additional bottom water data, particularly for DIC and TA, would be extremely useful in further validating the bottom water carbonate chemistry beyond the 2008-2010 analysis here.


## 3.2 Impact of forcing on linear trends

         The Bering10K BESTNPZ model has historically been utilized for a variety of fisheries management applications (Gibson and Spitz, 2011; Kearney et al., 2020). For these applications, the model hindcast timeframe needed to run through the present and extend back in time to cover

major transitions in the Bering Sea during the 1970s and 1980s. No individual forcing product provided this full timeframe, therefore, it was necessary to combine the CORE and CFSR forcing. Furthermore, the transition between products in 1995 was selected as the 1990s experienced relatively more stable climate variability for the Bering Sea, as this was after the shifts in the 1970s and 1980s, but prior to the temperature stanzas of the early 2000s (Stabeno et

al., 2012). However, any significant differences in either the atmospheric forcing or the oceanic boundary conditions between the datasets could generate a significant deviation in model results,





particularly immediately following the transition in 1995. Furthermore, this transition (i.e.
essentially a spin-up to the new model forcing) could generate erroneous linear trends when
calculated over the entire timeframe, that would represent a shift in the variable over a discrete

period, rather than a multi-decadal trend. To help clarify this potential influence, we ran a
separate simulation which branched off from the primary hindcast simulation in 1995 by
continuing the CORE forcing until 2003. We then compared these results to the primary
hindcast simulation (e.g. simulation that switches to CFSR in 1995) to assess the effect of this
transition in forcing.

Surface and bottom salinity for the Bering Sea shelf provides an example of how the shift
in forcing can generate an erroneous long-term trend. A noticeable decrease in salinity of ~ 0.5
psu immediately follows the switch to CFSR forcing and oceanic boundary conditions, which
does not occur when the CORE forcing and northeast Pacific model derived oceanic boundary
conditions are extended to 2003 (Fig. S1). This decrease generates a negative trend in surface

salinity when calculated over the entire timeframe, however, trends over the individual forcing
timeframes are extremely weak and of the opposite sign for the CORE timeframe.

To account for the potential influence of this transition in forcing, we report all timeseries
linear trends over three timeframes: 1.) the complete 1970-2022 CORE-CFS timeframe, 2.) the
1970-1994 CORE timeframe, and 3.) the 1998-2022 CFSR timeframe. We start the CFSR trends

in 1998 rather than 1995 to account for several years for the transition in forcing, based in part
on the re-equilibration to the new forcing by 1998 demonstrated in salinity (Fig. S1).
Furthermore, dividing the hindcast into the two timeframes of 1970-1994 and 1998-2022
produces two, equivalent 25-year time slices and will help elucidate any acceleration in trends.
Lastly, we show the results of the CORE simulation extended to 2003 for trend estimates in the

supplementary information, noting which variables exhibit consistent trends throughout both
forcing datasets, and which variables' long-term trends (estimated over 1970-2022) are impacted
by the forcing switch in 1995.

### 3.3 Bering Sea Shelf Acidification

Over the 1970-2022 model hindcast, annual surface and bottom $\Omega_{arag}$ and pH decrease,
while [$H^+$] increases for the Bering Sea shelf, with linear trends greater at the bottom compared



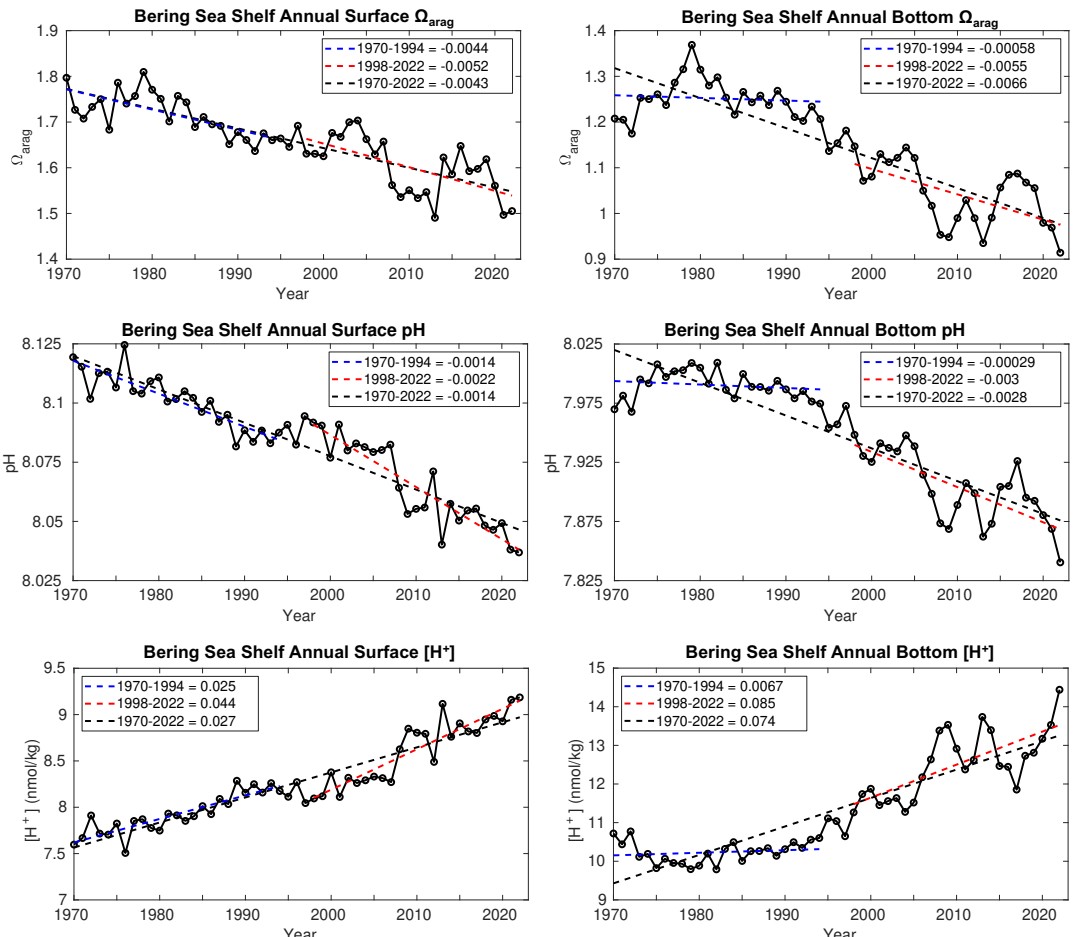

**Figure 6:** Timeseries plots of model annual average surface (left) and bottom (right) $\Omega_{arag}$ (top), pH (middle), and [H$^+$] bottom averaged over the Bering Sea shelf region. Also shown are the linear trend values over three different timeframes.

to the surface (Fig. 6). We show [H$^+$] in addition to pH because pH changes reflect relative [H$^+$]

changes and are, therefore, not ideal for comparisons between waters with different initial

chemistry conditions, such as between surface and bottom waters (Fassbender et al., 2017;

Fassbender et al., 2021). Surface $\Omega_{arag}$ ranges from 1.7-1.8 at the start of the simulation and

decreases to 1.5-1.6 by the end, surface pH ranges 8.1-8.125 and decreases to 8.025-8.05, and

surface [H$^+$] ranges from 7.5-7.75 nmol/kg at the start and increases to 9.25 nmol/kg by 2022.

Furthermore, the bottom pH trend from 1970-2022 is twice as great as the surface trend, while

the bottom [H$^+$] trend over the same timeframe is nearly three times as great as the surface [H$^+$]

trend. In fact, bottom acidity, as denoted by [H$^+$], increases by approximately 40% from 1970-

2022. These amplified bottom water carbonate trends are driven by the more recent timeframe,





as trends over the CORE-forced 1970-1994 timeframe are fairly weak, though these trends are a bit stronger when extending the CORE forcing to 2003 (Fig. S2). Surface trends in all three carbonate variables are comparable across all time frames, with slightly higher trends from 1998-2022 for pH and [H$^+$]. Notably, annual bottom $\Omega_{arag} < 1$ conditions first emerge in 2008, and

after 2020 stay below 1 for the remainder of the model simulation. Furthermore, bottom pH values are approaching 7.8 (e.g. conditions demonstrated to negatively affect growth and survival of red king crab) by the end of the model simulation.

Annual average surface $\Omega_{arag}$ and pH values from 1998-2022 are generally greater on the middle and outer shelf domains compared to the inner shelf domain (Fig. 7-8). Conversely,

bottom water values for both variables are generally greater for the inner shelf domain compared to the middle and outer shelf domains. The lowest bottom values tend to occur in the northwest Bering Sea shelf, in the Gulf of Anadyr. Relatively lower values of surface $\Omega_{arag}$ and pH are also apparent near the Yukon River delta. Most shelf surface waters have annual $\Omega_{arag} > 1.25$ and pH $\geq 8.0$. Bottom waters, however, are near or below the aragonite saturation horizon (i.e. $\Omega_{arag} = 1$)

for most of the middle and outer shelf, along with pH values $< 8.0$ and near 7.8 for the northwestern middle shelf domain. Surface $\Omega_{arag}$ and pH trends are spatially fairly consistent throughout the shelf, with slightly stronger, negative trends over the middle shelf and in the northwestern shelf near the Gulf of Anadyr (Fig. 7-8). Bottom water trends for both variables are more spatially heterogenous, with substantially greater trends on the outer shelf domain

compared to the rest of the shelf. This region, along with parts of the southeastern middle shelf domain, displays stronger, negative trends at the bottom compared to the surface, similar to the shelf-wide averaged timeseries plots in Fig. 6. [H$^+$] trends display similar spatial patterns as pH and are not shown here.

Vertical profiles of modeled pH at the M2 and M8 mooring locations highlight the onset

of seasonally occurring pH values $< 7.8$ (Fig. 9). At M2, these conditions do not occur in the hindcast until after 2005, at which point they seasonally occur somewhat regularly, and shoal to depths between 30-50m. At M8, pH $< 7.8$ waters rarely occur prior to 2000, after which they occur seasonally every year. Most years, these conditions also shoal to 30-50m, however, there are several years when they occur throughout the entire water column.





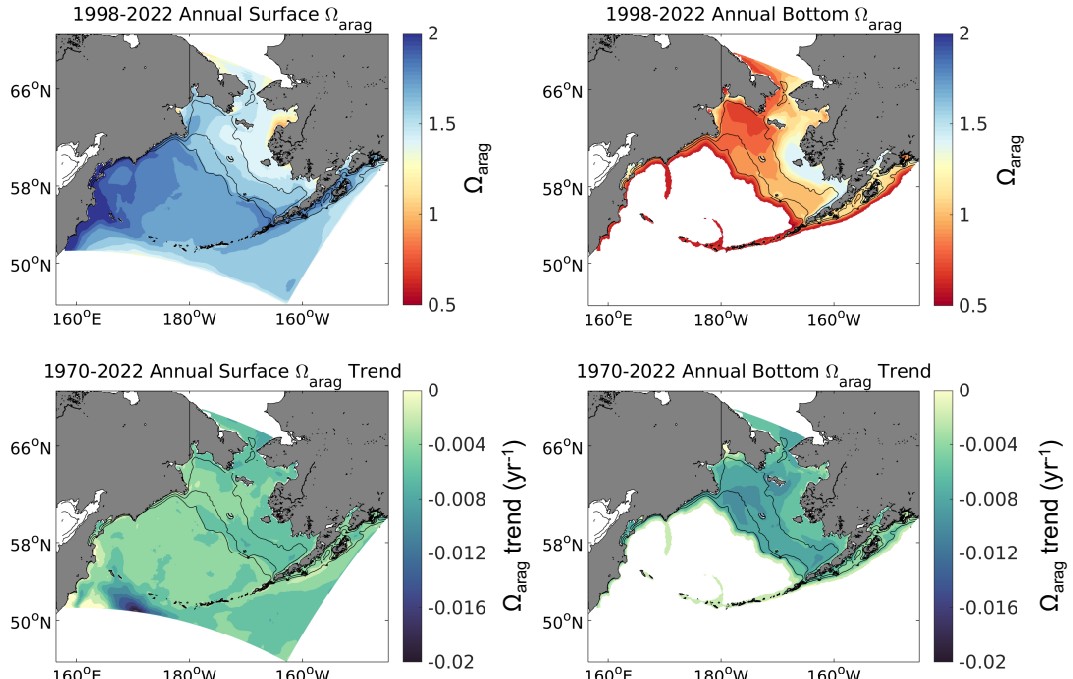

**Figure 7:** Spatial plots of model annual average surface (left) and bottom (right) $\Omega_{arag}$ form 1998-2022 (top) along with the linear trend for each grid cell (bottom) over the same timeframe. Bottom waters with depths > 1500m are omitted here as our focus is on the shelf.

### 3.4 Bering Sea Shelf Carbon Cycle

Atmospheric $CO_2$ concentrations significantly increase from 328 µatm in 1970 to 420 µatm by 2022, while the surface ocean $pCO_2$ for the Bering Sea shelf increases from 324 µatm in 1970 to 402 µatm in 2022 (Fig. 10a). This lag in the growth rate of surface ocean $pCO_2$ compared to the atmosphere generates a net decrease in D$pCO_2$ (i.e. $pCO_2^{ocean} - pCO_2^{atmo}$) and drives a more negative air-sea $CO_2$ flux, where a negative flux indicates a flux of carbon into the ocean (Fig. 10b, c). However, the more negative D$pCO_2$ values with greater carbon fluxes into the ocean tend to occur from 1995-2022, following the switch from CORE to CFSR forcing. Indeed, analysis of the CORE-extended hindcast indicates that the switch in forcing plays a significant role, with the CORE forcing suggesting higher oceanic surface $pCO_2$ values and more positive $CO_2$ flux values during the overlapping years (Fig. S3). Furthermore, while there is a negative trend in $CO_2$ flux over the entire 1970-2022 timeframe (under combined forcing), there is a very minimal negative trend over the 1970-2003 CORE forced timeframe and a slight positive trend over the 1998-2022 CFSR forced timeframe, indicating that the transition in



forcing is biasing the 1970-2022 trend (Fig. S3). To further illustrate this difference, we calculate the total carbon shelf sink using the spatial area of the shelf (i.e. area defined in Fig. 1;

804,393 km$^2$). For the CORE-forced 1970-1994 timeframe, the shelf was an annual carbon sink of 1.1 TgC/year, compared to an annual carbon sink of 7.9 TgC/year for the 1998-2022 CFSR-forced timeframe.

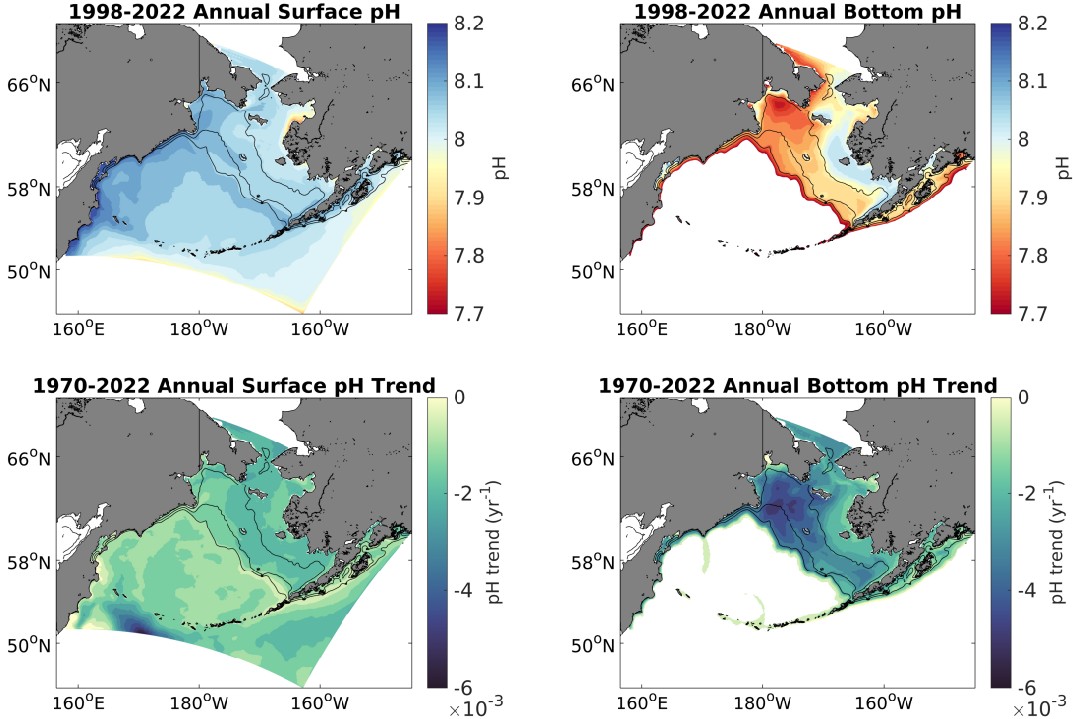

**Figure 8:** Spatial plots of model annual average surface (left) and bottom (right) pH form 1998-2022 (top) along with the linear trend for each grid cell (bottom) over the same timeframe. Bottom waters with depths > 1500m are omitted here as our focus is on the shelf.

Figure 11 illustrates that a substantial amount of this annual carbon uptake occurs within the middle and outer shelf domain and the northern Bering Sea inner shelf domain. Conversely, coastal waters near regions of significant riverine runoff (e.g. Yukon and Kuskokwim Rivers) are

an annual net carbon source. The spatial patterns of air-sea $CO_2$ flux are largely consistent with the spatial pattern in D$p$CO$_2$, though there are some areas where the two variables are not aligned (i.e. not the same sign). This is especially apparent for the off-shelf Bering Sea Basin, which displays slightly negative D$p$CO$_2$ values, but a relatively strong, positive (i.e. flux out of the





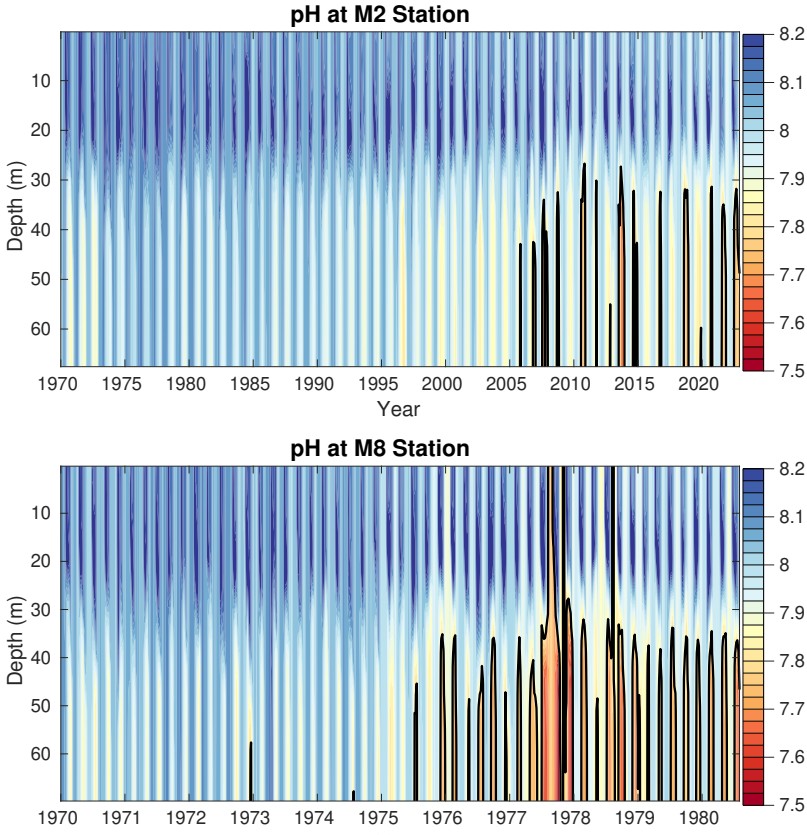

**Figure 9:** Model monthly averaged pH over the entire model timeseries at the M2 (top) and M8 (bottom) approximate model locations. The black contour line denotes the threshold for pH values < 7.8, which are conditions harmful to red king crab.

ocean) $CO_2$ flux. The difference in both variables between the CFSR and CORE forcing timeframes illustrates the substantial changes noted in Fig. 10. The off-shelf Bering Sea Basin in particular displays substantially greater magnitude, negative $DpCO_2$ and $CO_2$ flux values during the CFSR-forced timeframe. $CO_2$ flux values on the outer shelf domain and near the shelf-break are also substantially more negative (i.e. greater carbon uptake) during the CFSR-forced timeframe.

To further investigate the processes leading to the enhanced ocean carbon uptake, we examine the progression of the seasonal carbon cycle over each model decadal timeframe (Fig. 12). These figures reveal a non-uniform seasonal increase in surface ocean $pCO_2$, with the summer (May-September) values increasing at a much lower rate compared to the rest of the year. For example, the seasonal $pCO_2$ summer minimum increases by only 22 µatm over the

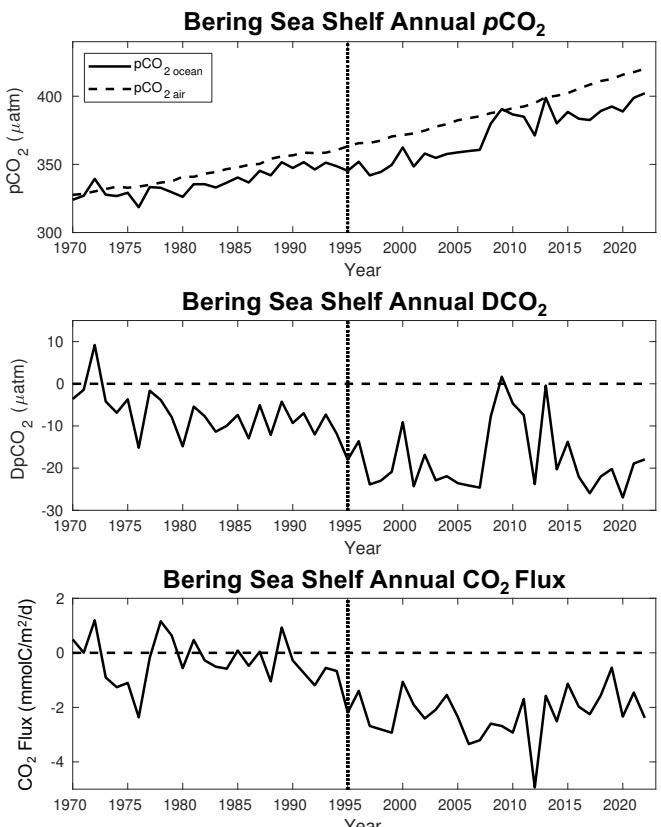

**Figure 10:** Timeseries of model annual average (top) surface ocean pCO₂ (black line) and atmospheric CO₂ concentration
(dashed line), DpCO₂ (middle), and CO₂ flux (bottom). Here, DpCO₂ is defined as $pCO_2^{ocean} - pCO_2^{atmo}$ and a negative CO₂ flux
signifies a flux of carbon into the ocean. The dotted line denotes the year where the forcing transitions from CORE to CFSR.

model timeframe, whereas the seasonal winter maximum in January increases by 93 µatm.

Atmospheric $pCO_2$ also increases over this timeframe, but with minimal changes in seasonality

(i.e. the seasonal amplitude increases by ~ 6 µatm over the entire timeframe). The overall effect

is a slight reduction in positive $CO_2$ flux (i.e. less carbon efflux to the atmosphere) during the

months when the shelf is a net source of carbon (November-March) but generates greater

magnitude, negative $DpCO_2$ and $CO_2$ flux values during the months when the shelf is a net

carbon sink (April-September). Notably, these enhanced negative $DpCO_2$ and $CO_2$ flux values

occur following the transition to CFSR-forcing.

To further understand changes in $pCO_2$, we separate the $pCO_2$ signal into a temperature

component and non-temperature component following Takahashi et al., (2002):





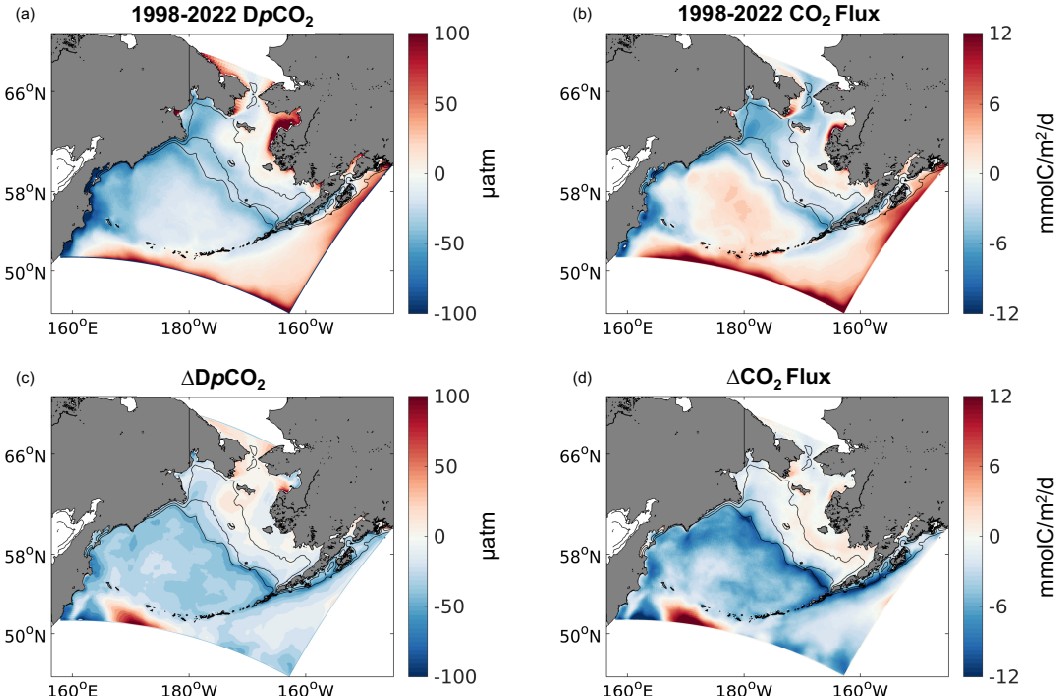

**Figure 11:** Spatial plots of model annual average surface (a) $DpCO_2$ and (b) $CO_2$ flux from 1998-2022. Also shown is (c) $\Delta DpCO_2$ and the (d) $\Delta CO_2$ flux calculated as the difference between the 1998-2022 and the 1970-1994 timeframes.

$$pCO_{2\,T} = \overline{pCO_2} * exp[0.0423(T - \bar{T})]$$

(5)

$$pCO_{2\,nonT} = pCO_2 * exp[0.0423(\bar{T} - T)]$$

(6)

where the overbars represent the model annual mean values, $pCO_{2\,T}$ is the temperature component reflecting the effect of thermal solubility on $pCO_2$, while $pCO_{2\,nonT}$ is the remaining $pCO_2$ effects governed by non-thermal components, including biological activity. Following equations 5 and 6, we can calculate the seasonal amplitude of both $pCO_{2\,T}$ and $pCO_{2\,nonT}$, which gives an indication of which component has a greater effect on determining the seasonal $pCO_2$. Figure 13 illustrates this comparison throughout the model timeframe. The seasonal amplitudes for both $pCO_{2\,T}$ and $pCO_{2\,nonT}$ increase over the model simulation, however, the amplitude for



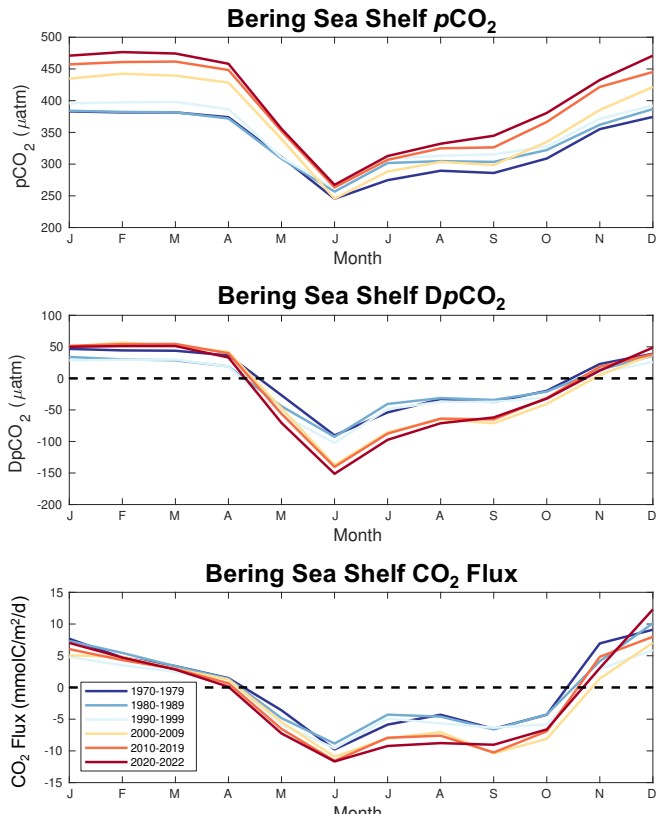

**Figure 12:** Seasonal plots of model surface ocean pCO2 (top), DpCO$_2$ (middle), and CO$_2$ flux (bottom) averaged over multiple timeframes.

$p\mathrm{CO}_{2\ \mathrm{nonT}}$ increases to a much greater extent. Furthermore, the $p\mathrm{CO}_{2\ \mathrm{nonT}}$ amplitude is always greater than the $p\mathrm{CO}_{2\ \mathrm{T}}$ amplitude, with the ratio increasing to greater than two.

Figure 6 illustrates that linear trends in $\Omega_{\mathrm{arag}}$ and pH are greater at the bottom compared to the surface, especially for the CFSR-forced timeframe. Figure 14 demonstrates that this is also true for the trend in DIC, where the bottom trend over the entire model hindcast is a little

over twice as strong compared to the surface. The CORE and CFSR forcing comparison illustrates that this enhanced bottom trend is a result of the CFSR-forced timeseries which is a factor of ~1.5 greater at the bottom compared to the surface for 1998-2022. Conversely, the CORE-forced surface trend is more than three times as strong as the bottom trend. However, extending the CORE forcing to 2003 doubles the bottom DIC trend, reducing this surface to

bottom trend comparison to less than a factor of two (Fig. S4). There are also positive trends in integrated primary production and bottom water remineralization, along with a negative trend in





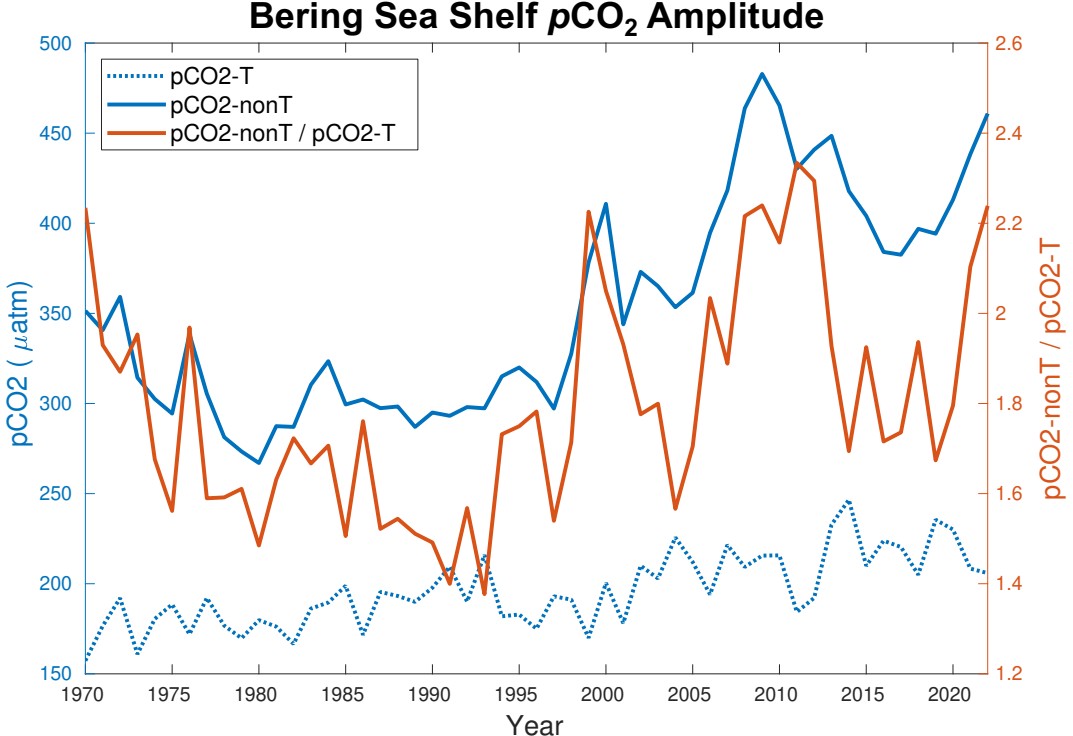

**Figure 13:** Timeseries of the yearly maximum seasonal amplitude of $pCO_2$-T (blue dotted line), $pCO_2$-nonT (solid blue line), and the ratio of $pCO_2$-nonT / $pCO_2$-T (orange line).

bottom oxygen concentrations over the entire model timeframe. Here, primary production refers to gross primary production (GPP) and remineralization encompasses all detrital remineralization and benthic excretion. Productivity and remineralization rates are both relatively high to start the model simulation, before decreasing to a minimum in the early 1990s, and then steadily increasing through the remainder of the model simulation. This leads to opposite trends in all

three variables between the CORE and CFSR forced timeframes, with CORE trending towards lower productivity, remineralization, and higher oxygen, but CFSR trending towards higher productivity, remineralization and lower oxygen. However, the CORE trends are more affected by the relatively anomalous initial values, and the extended CORE-forced simulation also suggests a shift towards higher productivity and remineralization, though not to the same extent

as the overlapping CFSR-forced years (Fig. S4). Over the entire model hindcast, productivity is strongly correlated with bottom remineralization (R = 0.92) and negatively correlated with bottom oxygen (R = -0.76).



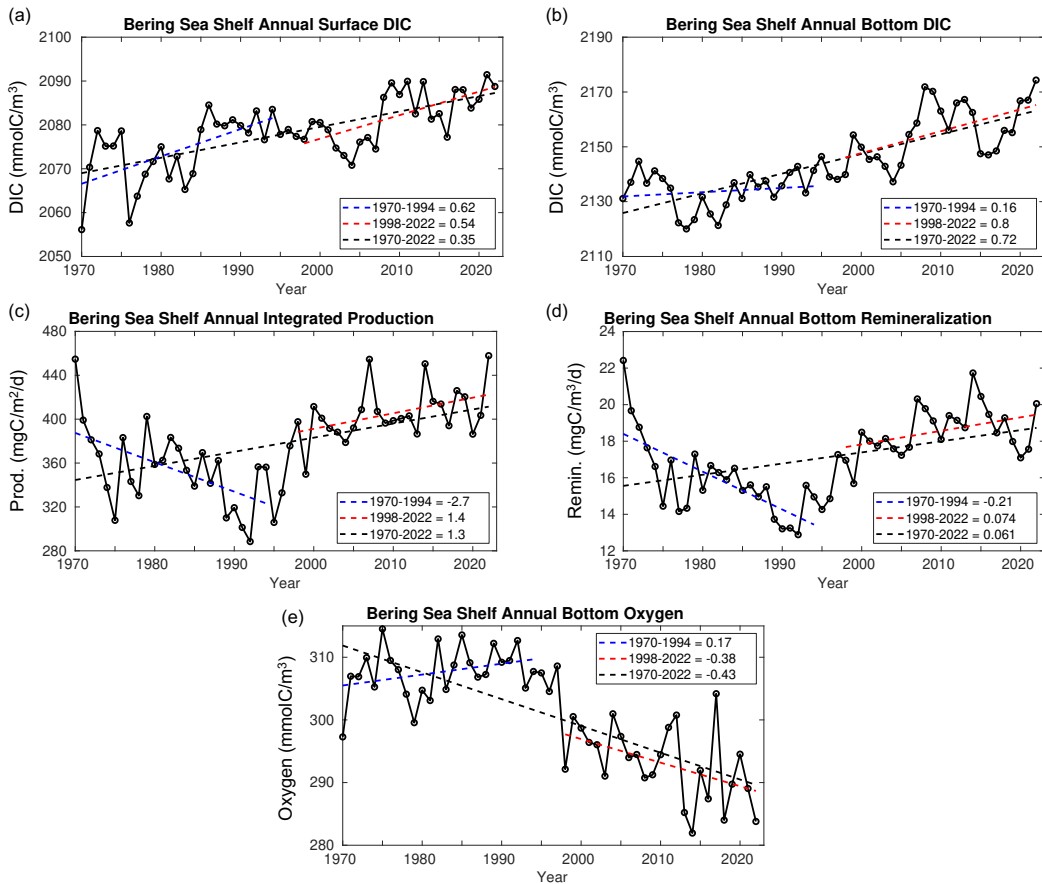

**Figure 14:** Timeseries plots of Bering Sea shelf model annual average (a) surface DIC, (b) bottom DIC, (c) depth integrated primary productivity, (d) bottom water remineralization, and (e) bottom water oxygen concentration. Also shown are the linear trend values over three different timeframes.

## 4 Discussion

Our model hindcast simulates surface $\Omega_{arag}$ and pH trends of -0.043 decade$^{-1}$ and -0.014 decade$^{-1}$ and bottom $\Omega_{arag}$ and pH trends of -0.066 decade and -0.028 decade$^{-1}$ respectively from 1970-2022 for the Bering Sea shelf. This surface pH trend is comparable to the global observed mean pH decline over a similar timeframe due to ocean acidification (Lauvset et al., 2015; Ma et al., 2023). Our surface $\Omega_{arag}$ trend is lower than the global observed $\Omega_{arag}$ trend of -0.071 decade$^{-1}$ (Ma et al., 2023), though the global high latitude trend is more comparable to our model trend. Pilcher et al., (2022) projected that surface $\Omega_{arag}$ on the Bering Sea shelf would decline by -0.044 to -0.097 decade$^{-1}$ from 2010-2100 under the RCP 4.5 and RCP 8.5 emissions scenarios





respectively, while surface pH would decline by -0.015 to -0.04 decade$^{-1}$. Thus, our hindcast simulation has a historical acidification rate from 1970-2022 that is comparable to the projected RCP 4.5 acidification rate. Conversely, the RCP 8.5 acidification rate is more than twice as great as our historical rate. This comparison provides context for the rate of change in carbonate chemistry that marine ecosystems have already experienced compared to the projected rate over the 21st Century.

Surface trends in $\Omega_{arag}$ are comparable across all model timeframes, while surface trends in pH and [H$^+$] are stronger over the last 25 years, reflecting a recent increase in the rate of acidification likely driven by the increased rate of atmospheric $CO_2$ growth. Interannual variability in surface carbonate variables also increased over the past 25 years, including the emergence of multi-year periods of sustained anomalous conditions. This is especially apparent for surface $\Omega_{arag}$, with periods of relatively high (e.g. 2001-2007 and 2014-2019) and low (e.g. 2008-2013) $\Omega_{arag}$ conditions. This coincides with the observed warm and cold temperature "stanzas" that have emerged for the Bering Sea shelf (Stabeno et al., 2012; Stabeno and Bell 2019). For the surface and bottom, warm temperatures lead to higher $\Omega_{arag}$ values while cold temperatures generate lower $\Omega_{arag}$ values. Pilcher et al., (2019) noted a similar phenomenon between a warm and cold temperature regime and attributed this to a combination of the thermal solubility effect on $\Omega_{arag}$ (i.e. cooling decreases $\Omega_{arag}$) and increased fall productivity and ocean carbon uptake. In our study, thermal solubility is likely also a contributor to recent $\Omega_{arag}$ variability; however, surface DIC (Fig. 14a) also displays a similar pattern between warm and cold temperature regimes suggesting the influence of changes in biogeochemistry (i.e. Pilcher et al., 2019). The warm and cold regimes also generate substantial differences in sea ice extent, which can impact the seasonal carbon cycle through changes in air-sea flux inhibition, the timing and composition of the spring phytoplankton bloom, and changes in the sea ice carbonate pump (e.g. Mortenson et al., 2020). A complete mechanistic breakdown of how the warm and cold temperatures regimes impact the seasonal carbon cycle and modify background OA rates is beyond the scope of this present manuscript but is the focus of planned future work.

The threat OA presents to Alaskan marine ecosystems demonstrates a clear need to develop accurate and reliable model-based OA products to support fisheries management. The recent emergence of multi-year anomalously low $\Omega_{arag}$ and pH conditions is significant because marine organisms may not be as resilient to longer cumulative exposure to acidic conditions



(Bednarsek et al., 2022). Furthermore, OA is gradually shifting waters to a lower $\Omega_{arag}$ and pH baseline and reduced buffer capacity, leading to a higher rate of extreme acidity events (Burger et al., 2020) and an amplification of the seasonal cycle (Kwiatkowski and Orr, 2018). It is therefore critical to track the development of high acidity water conditions on seasonal to annual timeframes to support tactical advice within the fisheries management process. To this end, we have developed an OA index for the Eastern Bering Sea shelf using annually updated output from our model hindcast (Fig. 15). This index indicates the area extent of the Bering Sea shelf where bottom waters are below threshold values of $\Omega_{arag}$ and pH from July-September. We specifically target summer bottom waters because this is when the seasonal bottom water respiration signal is greatest, thereby generating the most acidic seasonal conditions. The two biological thresholds are chosen as the aragonite saturation horizon, and a pH of 7.8, which has negative effects to red king and tanner crab growth and survival (Long et al., 2013a, b; Long et al., 2016). The spatial extent for both indices has greatly expanded over our model hindcast for both the entire Bering Sea shelf (Fig. 15a) and Bristol Bay (Fig. 15b), the location of a highly valuable red king crab fishery. Prior to 2005, between 5-10% of the shelf had pH < 7.8 conditions but by 2022 this jumped to more than 50% of the shelf spatial area. Thus, locations on the shelf that had rarely or never contained these conditions in our model hindcast prior to the early 2000s now regularly experience them (Fig. 9). Currently this index, along with spatial plots highlighting pH conditions on the shelf for the current year, are included in the annual NOAA Eastern Bering Sea Ecosystem Status Report (Siddon et al., 2022), a key report used by the North Pacific Fisheries Management Council for setting quotas.

Modeled bottom water acidification rates on the Bering Sea shelf are substantially greater compared to the surface, particularly for pH and [H+]. The bottom water amplified trends emerge over the past 25 years, coinciding with a net increase in primary productivity, and a subsequent increase in bottom water remineralization. The accumulation of anthropogenic carbon can also generate relatively greater changes in pH and [H+] in subsurface waters due to nonlinearities in the carbonate system (Fassbender et al., 2023), though anthropogenic carbon is not explicitly tracked in our model simulations. Our model results add to a growing body of literature suggesting that biological remineralization reduces water buffer capacity and can accelerate subsurface acidification rates (Cai et al., 2011; Feely et al., 2010; Cross et al., 2018; Kwiatkowski et al., 2020; Arroyo et al., 2022; Qi et al., 2022; Fassbender et al., 2023). Indeed,



Qi et al., (2022) found accelerated OA rates in the neighboring Chukchi Sea due to enhanced
subsurface biological remineralization. Previous observational studies have also noted a long-
term increase in primary productivity for both the Arctic Ocean (Lewis et al., 2020) and the
Bering Sea (Wang et al.,

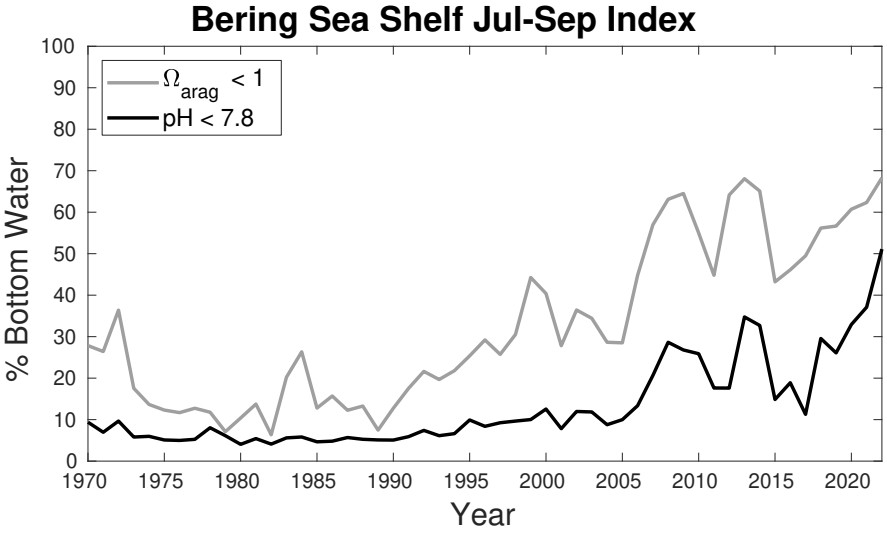

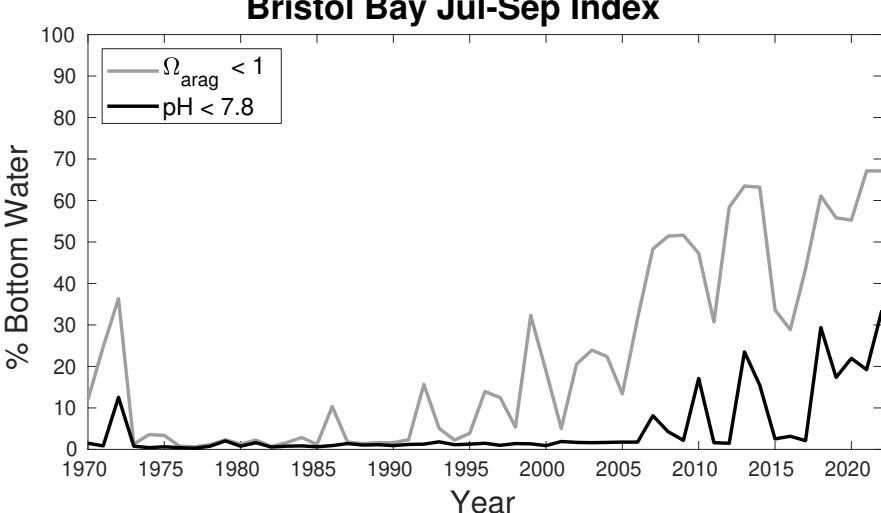

**Figure 15:** Timeseries plots of an OA indicator calculated as the spatial extent (i.e. percent of total area) of bottom waters with a
July-September $\Omega_{arag} < 1$ (grey line) and pH < 7.8 (black line). The total spatial are is the entire Bering Sea shelf for the top plot
and Bristol Bay for the bottom plot.

2022). Higher productivity in the Bering Sea has also been observed in warmer years (Lomas et

al., 2020), though model projections suggest that overall phytoplankton biomass will decrease





with future climate warming (Cheng et al., 2021). Thus, the enhanced productivity may be a
transient response to recent observed warming and sea ice decline and the resulting ongoing
ecological shift (Moore and Stabeno 2015; Overland et al., 2023). Interestingly, Pilcher et al.,
(2022) did not find accelerated bottom water acidification rates compared to the surface in their
projected OA rates for the Bering Sea shelf. These projections were generated using the same
Bering10K-BESTNPZ model presented in here, suggesting that the enhanced bottom OA rates in
our hindcast result from the model forcing.

 Here, we find that the Bering Sea shelf is an annual carbon sink of 1.1 – 7.9 TgC/year,
with the range resulting from the change in forcing between CORE and CFSR. Most of this
carbon uptake occurs on the middle and outer shelf domains, while the inner shelf domain
contains some regions of net carbon efflux, mostly located near river runoff. Previous estimates
for the shelf carbon sink have ranged from 2 – 67 TgC/year, and our estimate agrees with the 6.8
TgC/year estimate by Cross et al., (2013) that incorporated late fall/winter data when the shelf is
typically outgassing carbon. Notably, this range is significantly less than the previous model
estimate of 15-25 TgC/year by Pilcher et al., (2019), which was over a much shorter timeframe
(2003-2012) that only used the CFSR forcing (i.e. more comparable to our upper 7.9 TgC/year
estimate here). Using $pCO_2$ data from autonomous vehicles, Wang et al., (2022) found that
Bering Sea shelf carbon uptake has increased from 1989-2019 due to an increase in primary
productivity which suppressed summer $pCO_2$ values and generating more negative $DpCO_2$. Our
model results present a similar mechanism (Fig. 12) but are highly uncertain as this mechanism
appears to be sensitive to the switch in forcing. The substantial increase in the magnitude of the
$pCO_2$ $_{nonT}$ seasonal amplitude compared to $pCO_2$ $_T$ may also indicate that changes in productivity
and respiration are driving recent changes in the model carbon cycle and the amplified bottom
water acidification rates. However, anthropogenic carbon uptake can also generate large changes
in the $pCO_2$ $_{nonT}$ seasonal amplitude (Fassbender et al., 2018).

 Interestingly, the strongest model trends over the past 25 years are in the off-shelf Bering
Sea Basin. This region is a net annual source of carbon (Fig. 11), but the model suggests that
this carbon efflux has substantially declined over the past 25 years. This region also displays
divergent $DpCO_2$ and $CO_2$ flux patterns (i.e. negative $DpCO_2$ but positive $CO_2$ flux) on annual
timeframes, likely due to the influence of wind speed in determining the magnitude of the flux.
For example, wind speeds in the Bering Sea basin are much stronger in winter compared to





summer, thus positive winter efflux values will be greater in magnitude than negative summer influx values, generating a net positive annual average flux. However, our model results are likely more uncertain for this region because the substantially greater depths combined with our model terrain-following coordinates generate relatively deep surface grid cells, which may significantly influence the air-sea gas exchange.

A noted caveat to our model results is that the shift in atmospheric and boundary condition forcing in 1995 can lead to a shift in the system which impacts trends calculated over the entire model timeframe. For some model variables such as salinity and air-sea $CO_2$ flux, the impact is readily noticeable, particularly when extending the CORE forcing to 2003 (see supplementary information). Conversely, the extent to which this switch impacts the trends in 655   $\Omega_{arag}$ and pH are less clear. Surface $\Omega_{arag}$ and pH trends are largely consistent across all three timeframes, suggesting these trends are largely unaffected by the change in forcing. This result is not unexpected, given that surface acidification rates are strongly tied to the atmospheric $CO_2$ concentration, which is not impacted by the forcing shift. There is a moderate acceleration of the pH and $[H^+]$ trends over the last 25 years, however, the annual atmospheric $CO_2$ growth rate also 660  increases over this same timeframe. Meanwhile, bottom $\Omega_{arag}$ and pH display different trends over the CORE and CFSR timeframes, with essentially no trend with the former but steep negative trends with the latter. This result suggests that the 1970-2022 trend is not a product of a discontinuity created in 1995 by the change in forcing, but rather emerges over the 1998-2022 CFSR forcing. Thus, the accelerated bottom OA rates generated by the model may be dependent 665  on the CFSR forcing, as they are driven by enhanced productivity-remineralization that is not apparent in the CORE forced simulation. But it does not appear that these trends are artificially generated by the switch in forcing itself.

    It is also possible that these bottom water trends emerge over the more recent timeframe and are independent of the forcing, a conclusion supported by previous observational studies 670  (e.g. Qi et al., 2022; Wang et al., 2022). Indeed, extending the CORE forced simulation to 2003 generates a modest increase in bottom water acidification rates. Diagnosing the mechanism responsible for these differences in the forcing is beyond the scope of this manuscript, as our goal is rather to highlight which variables and trends are impacted by the transition in forcing. However, we note that the CORE atmospheric shortwave and longwave radiative forcing are 675  slightly adjusted to agree with the CFSR radiative forcing (Kearney et al., 2020) and that water





temperature comparisons between the two are comparable (Kearney 2021). Nonetheless, this study highlights the sensitivity of the simulated carbon cycle to small shifts in surface and boundary forcing and suggests that further constraints on the spinup and boundary condition forcing may be required as part of future model development.


## 5 Conclusions

We use a regional ocean biogeochemical model to simulate the Bering Sea shelf carbon cycle from 1970-2022. Over this timeframe, surface waters acidify at rates comparable to those observed in the global ocean, with a slight acceleration in the trend over the past 25 years. Shelf

bottom waters acidify at two to nearly three times the rate of surface waters, driven by increased productivity and subsurface respiration and remineralization. This mechanism leads to a substantial increase in the spatial extent of summer bottom waters with $\Omega_{arag} < 1$ and pH conditions harmful to red king crab, including parts of the shelf where these conditions previously did not occur during our model timeframe. To facilitate tracking these conditions and

support the fisheries management process, we have developed an OA index which is annually updated and presented as part of the NOAA Eastern Bering Sea Ecosystem Status Report. Lastly, we find that the Bering Sea shelf is an annual carbon sink of 1.1-7.9 TgC/year, which is lower than a previous model estimate of 15-25 TgC/year but is more consistent with the observational constraint of 6.8 TgC/year. The range in our estimate results from differences

between the two atmospheric forcing reanalysis products, with the higher estimate driven by relatively greater carbon uptake in summer and early fall and somewhat less winter carbon efflux.

## Code and data availability

The ROMS Bering10K model source code is available on Github here https://github.com/beringnpz/roms-bering-sea, and the model output is available on the PMEL THREDDS server through the Alaska Climate Integrated Modeling Project https://data.pmel.noaa.gov/aclim/thredds/catalog/files/B10K-K20P19_CORECFS.html. Atmospheric $CO_2$ values for Barrow and Mauna Loa are publicly available at the NOAA Earth

System Research Laboratories Global Monitoring Laboratory. M2 mooring $p$CO$_2$ data are



available at the NOAA National Centers for Environmental Information (NCEI)
https://www.ncei.noaa.gov/data/oceans/ncei/ocads/metadata/0157599.html.

**Author contributions**

DJP and JNC conceptualized the project and acquired the funding.  DJP ran the model
simulations and conducted the formal analysis, with assistance in model code development and
forcing generation from KAK, AJH, and WC. KAK, AJH, and WC provided data curation for
the model software and output.  JNC, NM, and LM assisted with the observational data and
methodology for model validation.  DJP generated the figures with assistance from LM.  JNC
and WC provided project administration support.  DJP prepared the manuscript, and all authors
commented on and contributed to the manuscript writing.

**Competing interests**

The authors declare that they have no conflict of interest.


**Acknowledgements**

This work was facilitated through the use of advanced computational, storage, and
networking infrastructure provided by the Hyak supercomputer system at the University of
Washington.  Stimulating conversations about the model output were also provided by our
colleagues at the UW Cooperative Institute for Climate, Ocean, and Ecosystem Studies, and the
PMEL Carbon and Eco-FOCI Groups.  Funding for this project was provided by the NOAA
Ocean Acidification Program (Research Organization Registry # 02bfn4816, NRDD # 20780)
through the Cooperative Institute for Climate, Ocean, and Ecosystem Studies under NOAA
Cooperative Agreement NA20OAR4320271.  This is CICOES contribution 2024-1354, PMEL
contribution 5619, EcoFOCI-1051, and PNNL-SA-197834.



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
