# Peer review of "Amplified bottom water acidification rates on the Bering Sea shelf from 1970-2022"

_EGUsphere, 2024_

## Author Comment (AC1)

Pilcher et al. used an ocean-biogeochemical model to investigate carbonate system variability in the Bering Sea, with a focus on the long-term trends of ocean acidification (OA) variables (Ω, pH, H+, pCO2), motivated by the need of relevant OA indices for marine resources management. They extended the temporal coverage of previous modeling studies, quantifying spatiotemporal trends during 1970-2022. Their model results showed a significant acceleration of the OA trend over the last 25-years. The simulated bottom trends are greater than the surface trends, presumably associated with an increased respiration/remineralization in response to enhanced phytoplankton production. This is an interesting and valuable study that contributes to better understand changes in the carbonate system in the Bering Sea, but I think additional work is required to clarify/improve the model settings and further explain the model results.

**RESPONSE: We thank the reviewer for the helpful feedback on our manuscript. We have conducted the additional requested analysis and have responded to the comments below. We think incorporating this feedback along with the comments from Reviewer 2 have greatly improved the manuscript.**

Main comments

1) Model forcing

The main results in this study relate to the long-term patterns in OA variables. A relevant question is therefore how robust the derived 1970-2022 trends are, and I have some concern about this. The author recognized potential issues associated with the use of multiple products to derive the surface model forcing (CORE for 1970-1994, NCEP-CFSR for 2011-2021, and NCEP-CFSv2 for 2011-2022) and boundary conditions of physical variables (Northeast Pacific model (NEP) for 1970-1994 and CFS for 1995-2022). Model results show that the CORE-to-CFS transition most likely altered the long-term patterns in the OA progression. Maybe I am missing something, but there are available atmospheric reanalysis products that cover the entire study period (e.g., ERA5: 1950-2022), so I am not clear why the authors decided to use those three different products. Could you clarify this?

**RESPONSE: We thank the reviewer for this insightful comment. We agree with their concerns regarding the shift in forcing from CORE to CFSR, hence why we made the decision to be explicitly clear on the timeframe over which the relevant trends develop. Our intention is to highlight that, while the shift from CORE to CFSR generates some artificial trends in certain variables (e.g. salinity), our primary result of enhanced bottom water OA rates compared to the surface is based on the CFSR forced simulation and is not artificially generated by the switch in forcing. Whether this result is constrained to just the CFSR forcing (as opposed to a different forcing product) is a separate issue (see next comment response). We chose to use the CORE and CFSR forcing because we have an extensive history of utilizing this forcing for Bering Sea ROMS model projects (e.g. Hermann et al., 2016; Pilcher et al., 2019; Kearney et al., 2020) and have found it to work quite well for the Bering Sea region. Furthermore, we are also in the process of testing 9-month seasonal forecasts, which utilize CFS for the**

retrospective seasonal reforecasts.  Bottom temperature seasonal forecasts have already been developed (e.g. Kearny et al., 2021) and we are now testing the biogeochemistry (Pilcher et al., in prep).  Because these reforecasts need to be skill assessed compared to a hindcast, using the CFSR hindcast is the best option.  Lastly, while there are some atmospheric reanalysis products that are available back to 1970, it is often more difficult to obtain an ocean reanalysis product back to that timeframe.  For example, while ERA5 starts in 1950, the corresponding ocean reanalysis product (ORA5) starts in 1979.  Thus, we would still have to combine multiple ocean reanalysis products to get the boundary conditions back to 1970, and our analysis of the salinity changes highlight that it's the transition between these boundary conditions that can induce the artificial trends.  We have added some additional text throughout the manuscript to help clarify.

An additional CORE-forced hindcast ending in 2003 was intended to clarify the CORE-CFS transition impact on the trends, but I do not think this extra analysis significantly helped to that goal. Instead of comparing if the CORE-CFS trends for 1970-2022 are like the CORE trends for 1970-2003, I would compare if trends derived from CORE and CFS forced experiments are consistent during the overlapping period of these two products. A comparison over the overlapping period could also help to clarify potential impacts on seasonal and interannual variability.

Maybe you could re-run the full hindcast using ERA5 as the only atmospheric forcing, so that the "forcing issue" would be limited to the NEP-to-CFS shift in the boundary conditions. In that case, you can run additional experiments to compare trends over the overlapping periods of NEP and CFS.

RESPONSE:  Similar to the previous comment, our goal with this comparison is not to suggest that trends between the CORE and CFSR forced products are the same.  Forcing products are imperfect and contain uncertainty which will impact our model results (e.g. Jung et al., 2014 doi:10.1002/2013GL059040; Lima et al., 2018 doi:10.1029/2018JC013919).  Rather, our goal is to illustrate that our longterm trends are not artificially generated by the switch in forcing. A separate hindcast utilizing the ERA5 forcing for the entire timeframe is an interesting project idea, but this would be a substantial additional effort, that would then shift the focus of the project towards understanding model sensitivity to the forcing (both atmospheric and horizontal boundary conditions), which is not our goal here.

2) Salinity trend shift

Figure S1 show a strong trend shift in salinity associated with the change in the model forcing products (CORE to CFS).  I suggest reporting the mean alkalinity series at surface and bottom as supplementary figure, since salinity and alkalinity are usually strongly correlated. Did you get a similar trend change in alkalinity? Since salinity and alkalinity are drivers of $\Omega$, pH, and pCO2, then that shift could significantly impact all the reported OA trends. Could you discuss about it?

RESPONSE: This is a great suggestion, and we have added the alkalinity timeseries to the

supplement, along with the change in the DIC and TA boundary conditions that result from the salinity shift, following the recommendation of Reviewer 2.  Similar to salinity, there is an overall decreasing trend in alkalinity, which appears to be largely related to the forcing shift, although there is still a slight decreasing trend over just the CFS timeframe.  While this will impact the carbonate chemistry, the decrease in salinity will also decrease DIC, thus mitigating some of the effect.  Additionally, the change in salinity from the forcing can modify the incoming boundary conditions, which have a relatively greater effect on DIC than TA due to the empirical relationship (i.e. comment raised by Reviewer 2).  We also further expand on the drivers based on the Taylor series decomposition mentioned further below.  We have expanded the following text in the manuscript to further describe these details and added a new supplemental figure.  We also show below a figure for the TA trend in the 3 ESMs for comparison:

*"The lateral boundary conditions for DIC and TA are calculated via linear regressions with salinity through the following equations below, derived from observational data collected primarily from 2008-2010 (Pilcher et al., 2019).*

$$S < 32.6 \; DIC = 58.5 * S + 191.2 + \Delta DIC(t)^{atmo} \qquad (1)$$
$$S \geq 32.6 \; DIC = 140.4 * S - 2478.7 + \Delta DIC(t)^{atmo} \qquad (2)$$
$$S < 33.6 \; TA = 49.6 * S + 600.6 \qquad (3)$$
$$S \geq 33.6 \; TA = 141.8 * S - 2494.4 \qquad (4)$$

*The salinity-DIC regression has changed over time as the oceanic uptake of $CO_2$ has increased the DIC concentration of waters, with no effect on salinity.  Thus, using this same relationship for the boundary conditions at the start of the hindcast in 1970 would artificially increase DIC. To account for changes in DIC over time, we center the DIC-salinity relationship on the year 2009 (i.e. midpoint of 2008-2010 sampling timeframe) and subtract (add) DIC for years before (after) 2009.  The DIC value added or subtracted ($\Delta DIC^{atmo}$ in equations 1-2) for year(t) is obtained from the linear trend in DIC (Fig. S1) calculated from the historical runs of the Coupled Model Intercomparison Phase 6 (CMIP6) over the 1970-2009 timeframe from the mean of three different Earth System Models (GFDL-ESM4, CESM2, and MIROC-ES2L).  These three ESMs were selected as they have been used previously in the Bering10K regional dynamical downscaling (Cheng et al., 2021; Pilcher et al., 2022).  We chose to use this method to gain the higher spatial resolution, particularly in the vertical, provided by the ESM output. We only use the DIC trend from the CMIP6 ESMs and omit any TA trend because the TA trends over this timeframe are much smaller and are tied to changes in salinity (Hinrichs et al., 2023), which is accounted for in our salinity-TA relationship at the boundary."*

[Figure]

3) pCO2 patterns

There are not many observations during fall-winter, but the M2 records in 2021 suggests that the model is overestimating pCO2 during those seasons. If that is correct, then you could conclude that the model has an overall positive bias in pCO2 (and negative bias in pH), which could have a strong impact on the air-sea CO2 fluxes. You could discuss about it and tone done your results related to CO2 fluxes.

**RESPONSE: We agree that the M2 comparison does suggest an overall positive bias in model pCO$_2$. As noted in the text, this bias appears to evolve from an underestimation in the spring pCO$_2$ drawdown, which leads to subsequent overestimations of pCO$_2$ throughout the summer. However, it should be noted that M2 is just a single point in the model domain, and we suggest caution when extrapolating a M2 bias to an overall model bias. For example, the Saildrone comparisons (Figure 5), which provide a much larger spatial footprint, also suggest some positive pCO$_2$ biases in early spring, but there are also negative biases apparent in the southward transect in 2019. A property-property comparison of all of these Saildrone observations, suggests a very small overall bias (Figure below). We lastly note that our total annual carbon flux values of 1.1-7.9 TgC/year compare reasonably well with the broad observational estimates of 2-67 TgC/year, and specifically the 6.8 TgC/year from Cross et al., 2013, which incorporates winter flux estimates missing from the earlier studies.**

[Figure]

4) Underlying drivers

A Taylor series decomposition could provide valuable insights about the underlying drivers of OA variability, helping to identify the causes for the carbonate system trend changes, support the hypothesis of a biological driven increase in the bottom OA trends, and identify if salinity and alkalinity play any role on the OA progression changes. Although the authors mentioned  that diagnostic mechanisms are beyond the scope of this study, I strongly recommend adding a Taylor decomposition analysis, especially considering that a paper describing historical OA pattern was already published (Pilcher et al., 2019).

**RESPONSE: This is an excellent suggestion by the reviewer, and we have now included a new figure along with the accompanying text:**

*"To further understand the drivers behind changes in the carbonate chemistry, we also use a first-order Taylor series to decompose changes in $pCO_2$, $\Omega_{arag}$, and $[H^+]$ into the four primary drivers:*

$$\Delta\phi = \frac{\partial\phi}{\partial DIC}\Delta DIC + \frac{\partial\phi}{\partial TA}\Delta TA + \frac{\partial\phi}{\partial Salt}\Delta Salt + \frac{\partial\phi}{\partial Temp}\Delta Temp \qquad (11)$$

*Where $\Delta\phi$ represents the time change in the calculated carbonate parameter ($pCO_2$, $\Omega_{arag}$, or $[H^+]$), and the four variables on the right-hand side of the equation account for the contributions of DIC, TA, salinity, and temperature respectively.  The partial derivatives are calculated through small perturbations using CO2SYS (Lewis and Wallace, 1998; Sharp et al., 2023).  We employ the Taylor series decomposition for both the entire 1970-2022 timeframe, in addition to the CFSR 1998-2022 timeframe (Fig. 15).  This decomposition further highlights that the OA trends are driven by increasing DIC, particularly for bottom waters.  Surface carbonate trends are also driven to a lesser extent by decreasing TA over the 1970-2022 timeframe, though this effect is somewhat diminished during the more recent 1998-2022.  On this timeframe, warming temperatures emerge as a driver for surface and bottom $pCO_2$ and $[H^+]$, though still lower in magnitude than DIC."*

[Figure]

**Figure 15: Taylor series decomposition of trends in $pCO_2$ (top), $\Omega_{arag}$ (middle), and [H+] (bottom) for surface and bottom waters over the 1970-2022 and 1998-2022 timeframes.**

Minor comments:

I suggest tone done all the modeling results, considering that an ocean-BGC model rather suggest than demonstrate the OA patterns. For example, in the abstract: "surface Ωarag decreases by -0.043 decade-1 and surface pH by -0.014 decade-1" => "model results suggest that Ωarag decreases by 0.043 decade-1 and surface pH by -0.014 decade-1"

**RESPONSE: Thank you for the suggestion, we have toned down some of the language for underlying results through the text and the abstract.**

168-171: Could you provide the spatial resolution of the Northeast Pacific model and the CFS products used for the boundary conditions?

**RESPONSE: We have added the spatial resolutions for both models.**

174: How did you estimate the empirical climatological profiles for iron?

**RESPONSE: We have added the following description to the text**

*"Water column iron concentrations are nudged towards empirical climatological profiles, which use an analytical function based on Seward line data in the Gulf of Alaska for coastal regions (Hinckley et. al, 2009).  On-shelf values are set to 2.0 mmol/$m^3$ at the surface and 4.0 mmol/$m^3$ at depth, and this gradient transitions linearly to 0.01 mmol/$m^3$ at the surface and 2 mmol/$m^3$ at depth in water depths greater than 100m."*

393: "variables are comparable" what do you mean?

**RESPONSE: That the trends for all three variables at the surface are similar.  For clarity, we have changed to "similar" rather than comparable.**

415: "seasonally occurring" => provide the season name.

**RESPONSE: Here, seasonally was not referring to a specific season.  We have removed for clarity.**

455: I suggest using the same x-axis range (1970-2022) for the two panels, independently that the M8 station data extend only until the 80s.

**RESPONSE: Thank you for catching this, the x-axis was mislabeled as it does indeed extend from 1970-2022.  We have corrected the axis label.**

---

## Author Comment (AC2)

This study present a long term hindcast (1970 to 2022) of the carbonate system on the Bering Sea shelf. Such long term hindcast are very useful to understand the impact of environmental changes on the ecosystem and are very useful for fisheries management. The model results show an acidification trend, which is more important at the bottom than at the surface. The trends of the different OA variables are reasonable and within the range observed elsewhere. However, there is no atmospheric forcing data that covers the whole period of the hindcast and thus 3 different products were used. This approach seems to introduce an artificial trend when the forcing is switched from one product to the other. This issue is well discussed in the manuscript but unfortunately it creates uncertainty in the results. I think a little more could be done to understand what is going on when moving from the CORE forcing to the CFSR forcing. It would be very useful to know how the temperature and salinity change at the open boundaries of the model, is it why a general decrease in salinity is observed? How is the salinity-DIC relationship modified, does this lead to more or less DIC flowing into the system. Water from which depth at the boundary will affect the Bering Sea shelf? What is the impact of not modifying the TA relationship at the boundary? Some processes affecting DIC (not the air-sea flux), water mass changes, can also slightly affect TA and thus the relationship. Modifying one variable but not the other could introduce a bias in pH and omega. Was this explored with the Earth System Models or only dismissed? And finally, the large trend towards the end of the hindcast was attributed to an increase in primary production. Could you explain why PP is increasing? The salinity gradient between the surface and the bottom seems smaller with the CFSR forcing, is there a decrease in stratification that leads to a greater nutrient availability? Or is there more light available? How does the PAR forcing (SW, clouds) compare between the different products? Is it a temperature effect (no info is given on the temperature change)?

To summarize, I think it is a very useful study but giving a little more information on the differences between the different forcing, and some explanations on the causes of the observed changes (i.e., salinity, primary production) would help build confidence in the results. The manuscript is very well written and clear. I think once some of the above questions are answered and delt with if necessary, the manuscript would meet all the criteria for publication in this journal.

**RESPONSE:** We thank the reviewer for these insightful comments and positive feedback on our manuscript and have further commented on the points raised below. Overall, we agree with many of the points raised and have furthered our analysis and added some additional text to help clarify the concerns. Specifically, we note that there was confusion regarding how the DIC and TA boundary conditions were generated, which was also reflected by Reviewer 1. Thus, we've added some additional details to help clarify and also now discuss the effect of the shift in salinity from the model forcing on our DIC and TA boundary conditions. Furthermore, we have added some additional discussion concerning the effect that the shift in forcing has on the underlying model trends, which was a concern also raised by Reviewer 1. Lastly, we have included some additional analysis on the mechanisms for the increase in primary productivity within the model. Detailed descriptions of these points are described below under the specific comments.

Specific and technical comments :
Line 58 : Could refer to Figure 1

**RESPONSE: Great suggestion, we now refer here to Fig. 1**

Line 107: lower pH values -> not clear lower than what. It would be better to actually put the pH values.

**RESPONSE: We now specifically note lower than pH of 7.8**

Line 110: Seung and Punt not in the reference list.

**RESPONSE: Thank you for catching this, missing references are now included.**

Line 111: on the Bering …

**RESPONSE: Changed "to" to "on".**

Line 112: some discussion on the role of OA: who had these discussions, any ref to a workshop or something like that?

**RESPONSE: Reference included.**

Line 125: could you specify what was the length of the previous hindcast?

**RESPONSE: Previous hindcast timeframe of 2002-2012 now included.**

Line 147: add parenthesis (2020).

**RESPONSE: Added**

Line 160: How many rivers are there in the model? How many rivers have discharge and DIC, TA values? The same values are added to all the rivers that do not have measurements (if that is the case)? What do the TA and DIC values look like (i.e. range of values). What are the largest rivers in term of runoff. I think some more info would be useful especially that the river data seem to have a large impact in some regions of the Bering Sea.

**RESPONSE: This is a great suggestion and we have expanded this section as follows:**

*"Riverine freshwater runoff flux is prescribed following freshwater discharge data from 28 watersheds in Alaska and Russia, including the Yukon River which supplies roughly 50% of the total freshwater flux to the Bering Sea shelf (Kearney, 2019).  This river runoff contains seasonally varying concentrations of DIC (1480-4100 µmol/kg) and TA (1238-2743 µmol/kg)*

*following data collected at Pilot Station at the mouth of the Yukon River (Striegl et al., 2007; PARTNERS, 2010, Pilcher et al., 2019)."*

Line 166: 1995-2010 instead of 2011? Why didn't you use the CFSv2 reforecast for your whole 1995-2021 period? These 3 different forcing actually creates 2 transitions.

**RESPONSE: This may be somewhat of a semantics issue. The "Climate Forecast System" provides at least 5 different data products, two of which are the CFS[v1] Reanalysis and CFSv2 Operational Analysis. The latter uses updated versions of the CFS model components and assimilation framework, and also began incorporating near-real-time data. But there is no overlap between the two (i.e., there is no CFSRv2 or CFSv1 operational analysis), just a switch in March 2011.**

Lines 175 and following: The adjustments made to the relationship and the resulting DIC over the full salinity range is not clear to me. I think a figure would help. Also could you name the 3 ESMs used? In the earth system models the DIC changes include more than the results of atmospheric CO2 influx and other processes could impact TA as well. So at depth it might be relevant to include the changes in TA as well (from the ESMs) to preserve the TA-DIC balance.

**RESPONSE: We have expanded the description of our lateral boundary conditions and have included the equations to help clarify how these values are generated. The reviewer is correct that TA also changes over the historical timeframe of the ESM, however, these changes are very small (figure below). Furthermore, part of these changes will be driven by changes in salinity. Because our salinity boundary conditions are tied to the CFS forcing, we don't want to decouple changes in salinity-TA by incorporating TA output from the ESM, but salinity changes from the forcing product. Thus, we decided to not include this relatively small effect for model simplicity and to minimize additional external forcings that can add further uncertainty. DIC was treated differently, since the trend from atmospheric $CO_2$ is significant over this timeframe. We have modified and added to the manuscript text as follows below:**

*"The lateral boundary conditions for DIC and TA are calculated via linear regressions with salinity through the following equations below, derived from observational data collected primarily from 2008-2010 (Pilcher et al., 2019).*

$$S < 32.6 \; DIC = 58.5 * S + 191.2 + \Delta DIC(t)^{atmo} \qquad (1)$$
$$S \geq 32.6 \; DIC = 140.4 * S - 2478.7 + \Delta DIC(t)^{atmo} \qquad (2)$$
$$S < 33.6 \; TA = 49.6 * S + 600.6 \qquad (3)$$
$$S \geq 33.6 \; TA = 141.8 * S - 2494.4 \qquad (4)$$

*The salinity-DIC regression has changed over time as the oceanic uptake of $CO_2$ has increased the DIC concentration of waters, with no effect on salinity. Thus, using this same relationship for the boundary conditions at the start of the hindcast in 1970 would introduce a high DIC bias. To account for changes in DIC over time, we center the DIC-salinity relationship on the year 2009 (i.e. midpoint of 2008-2010 sampling timeframe) and subtract (add) DIC for years*

*before (after) 2009. The DIC value added or subtracted (ΔDIC^{atmo} in equations 1-2) for year(t) is calculated from the linear trend in DIC (Fig. S1) calculated from the historical runs of the Coupled Model Intercomparison Phase 6 (CMIP6) over the 1970-2009 timeframe from the mean of three different Earth System Models (GFDL-ESM4, CESM2, and MIROC-ES2L). These three ESMs were selected as they have been coupled previously with the Bering10K regional model (Cheng et al., 2021; Pilcher et al., 2022). We chose to use this method to gain the higher spatial resolution, particularly in the vertical, provided by the ESM output. We only use the DIC trend from the CMIP6 ESMs and omit any TA trend because the TA trends over this timeframe are much smaller and are tied to changes in salinity (Hinrichs et al., 2023), which is accounted for in our salinity-TA relationship at the boundary."*

[Figure]

**Figure: Ensemble mean annual trends in DIC and TA from the 3 Earth System Models, downscaled to our west and south boundary conditions. The West and South boundaries on the respective η-axis and ξ-axis are shown in Figure 1.**

Line 120. Wd should be in italics

**RESPONSE: Change made**

Line 265: How are the off shelf break water affecting shelf water? A related question is how are the model boundary conditions affecting the shelf water, i.e. only the properties at similar depths (down to 200 m) or deeper?

**RESPONSE: We have added the following text and reference:**

**"*Modeled transport across the shelf break is relatively small, with most on-shelf water arriving through the Aleutian Islands, with shelf water residence times generally less than 3 years (Mordy et al., 2021).*"**

Line 282: Why is omega more variable?

**RESPONSE: Good question, we think it may be due to underlying complexities and nonlinearity of the marine carbonate system. It's the only variable with lower variability than the observations, which is apparent both in this Target diagram and when comparing the standard deviations between the model and the observations. For the observations to have higher variability in temperature, salinity, DIC, and total alkalinity, but the model to have higher variability in $\Omega_{arag}$ (which is a function of those 4 variables), may indicate that some of the model-data differences are combining in a way that amplifies variability in the model. The fact that this is not the case for pH may suggest that temperature is playing a role, since temperature has a relatively greater effect on $\Omega_{arag}$ than pH.**

Figure 4: There is a smaller peak that precedes the larger peak in the model. What is it caused by? In the observation we rather see a slowdown of the increase.

**RESPONSE: We think the reviewer here is referring to the gradual $pCO_2$ increase in summer (following the sharp drawdown from the spring bloom), which in some years appears as a peak due to a smaller magnitude $pCO_2$ drawdown in fall (e.g. 2020), though not always (e.g. 2018). The modeled smaller drawdown in fall is from a fall phytoplankton bloom that often develops as stratification breaks down and there is a pulse of nutrients into the euphotic zone, while light and temperature conditions are still sufficient. This bloom is relatively short-lived though, before late fall mixing brings subsurface carbon back to the surface layer, generating relatively high $pCO_2$ values for both the model, and the mooring when it is deployed late enough into the year.**

Line 330: Would the TA and DIC bias originate from your treatment at the open boundary (i.e, increasing DIC and not TA).

**RESPONSE: Biases arising from lateral boundary conditions are certainly one possibility, but it's unlikely that our relatively small biases in DIC and TA are related specifically to incorporating a trend in DIC but not TA. First, the model data comparison is over 2008-2010, which is when the DIC trend is near 0. This is because the salinity-DIC empirical fit used these same observational data points, thus any accumulation of atmospheric $CO_2$ is accounted for, centered on 2009. Second, the slight trend in TA from the ESMs is negative, therefore, incorporating this would only add to the negative model TA bias.**

Line 356: This decrease in salinity will lead to important changes in the DIC at your open boundary (?) and thus to the important change in $H^+$, pH and omega on the bottom? This comes back to my previous comments about the usefulness of actually displaying the changes in the S-DIC relationship and mentioning the importance of the OBC on the shelf conditions.

**RESPONSE:** This is a really great point raised by the reviewer that we originally overlooked in our description of the salinity decrease. We expect this change in salinity to decrease DIC, but also TA, thus the effects on carbonate variables will be more muted, since both values are simultaneously decreasing (also see comment in response to Reviewer 1). We now include an additional figure in the supplement which illustrates the change in the open boundary conditions for DIC and TA that directly results from this change in salinity. This comparison illustrates that surface DIC and TA changes at the open boundary are relatively small (mostly 10-20 mmol/m$^3$) and of the same sign. There are some higher magnitude differences in intermediate waters, particularly for the southern boundary, where a relatively large decrease in salinity leads to larger changes in DIC (80-130 mmol/m$^3$) and TA (40-70 mmol/m$^3$). At our mean shelf values of temperature (3.5°C), salinity (32 psu), DIC (2078 mmol/m$^3$), and TA (2224 mmol/m$^3$), we would expect this change to yield a pH and $\Omega_{arag}$ increase of 0.11-0.16 and 0.34-0.50, respectively. Thus, these changes generated by the decrease in salinity would work to counteract the ocean acidification trends in the model, particularly the subsurface trends (where we see the highest OA reates). We have included additional explanation in the text as excerpted below:

*"This shift in salinity will also impact DIC and total alkalinity through the salinity regression equations used to calculate the horizontal open boundary conditions (Equations 1-4). This leads to a decrease in DIC and total alkalinity within the open boundary conditions that is greater in magnitude in intermediate waters (Fig. S3). The effect is greater on DIC relative to TA due differences in the regression equations. The net effect on shelf-wide conditions is readily apparent for total alkalinity (Fig. S4), but is more muted with DIC, likely due to the relatively stronger effect of biology and air-sea gas exchange."*

[Figure]

**Figure S3: Changes in the western and eastern boundary for salinity (left column), DIC (middle), and TA (right) that result from the shift in forcing from CORE to CFSR in 1995. Delta values are specifically calculated between the two timeframes of 1985-1994 and 1995-2004. The West and South boundaries on the respective η-axis and ξ-axis are shown in Figure 1.**

Line 397: Could you add a reference?

**RESPONSE: References added.**

Line 417: Wrong x axis at M8

**RESPONSE: Thank you for catching this, we have corrected the axis label.**

Line 460-461: Could you say why? What are the changes in temperature between the two forcings?

**RESPONSE: The difference in D$p$CO$_2$ between the two forcings generates the difference in CO$_2$ flux. That difference in D$p$CO$_2$ is likely not driven by temperature as temperature trends throughout the model timeseries are fairly minimal. This is also apparent in the Taylor series decomposition recommended by Reviewer 1, where temperature is a minimal driver over the entire timeseries, and a slight, positive driver (i.e. warming) over the CFS timeframe. Primary production is greater in the outer shelf during CFS, so this may be partly the driver of the more negative D$p$CO2, but it also could be a product of changes in shelf mixing or incoming waters from the Alaska Coastal Current.**

Figure 10: p missing in DpCO2 panel title.

**RESPONSE: Figure panel title changed**

Line 510: Why is primary production increasing with the CFSR forcing? Is there a change in stratification (the difference is surface and bottom salinities seems to suggests that) that would provide more nutrients to the upper layer? Any changes in PAR from atmospheric forcing?

**RESPONSE: This is a great question and there are two parts to the answer. First, the increase in primary production between the CORE and CFSR forcing is tied to an increase in nutrient concentrations. However, the CFSR forcing also displays somewhat reduced shortwave radiation, but there is a positive trend in shortwave radiative forcing during the CFSR timeframe. This is likely due to the impact of relatively high, late sea ice extent during the cold years (e.g. 2008-2012), compared to the opposite during the warm years (e.g. 2017-2019). Mixed layer depths increase slightly throughout the CFSR timeframe, but the overall trends are fairly low. We have added some additional clarifying text to the manuscript below.**

*"This increase in productivity is tied to an increase in nitrate concentrations from the CORE to CFSR forcing, along with a positive trend in shortwave radiative forcing during the CFSR forced timeframe. "*

[Figure]

[Figure]

Line 511: Any oxygen observations that would support the remineralization rates at the bottom?

**RESPONSE: There are some bottom water observations that have been collected by the Ecosystems & Fisheries-Oceanography Coordinated Investigations (EcoFOCI) program, but unfortunately these discrete measurements are not of sufficient temporal resolution to identify any long-term changes, especially given the relatively high recent natural variability. However, EcoFOCI has deployed a moored high-resolution Profiling Crawler over the past several years which captures depth-resolved oxygen at very high temporal frequency (Nielsen et al., 2023, doi:10.1029/2022JC019076). With continued future deployments, it's possible that this dataset will soon acquire the temporal resolution necessary to resolve any long-term trend.**

Figure 15: replace are by area in legend

**RESPONSE: Change made**

**Citation**: https://doi.org/10.5194/egusphere-2024-1096-RC2

---

## Author Response (AR2)

Reviewer 1 Comments

The authors addressed most of my comments. However, I still consider that the change in the forcing product (CORE to CFS) could explain (to some degree) the enhanced OA over the last decades. I understand that running a new hindcast using a single data product for the forcing implies significant additional work, and is beyond the scope of this article. Under this limitation, the forcing issue should be clearly identified as a caveat that adds uncertainty to the derived long-term trends. My suggestion is tone down the results a bit in the discussion.

**RESPONSE: We thank the reviewer for the helpful comments. We agree that the forcing switch adds uncertainty, and indeed have a full paragraph in the discussion (lines 712-729) highlighting this as a caveat to our model results. We also agree that re-running the entire hindcast with a new forcing product is a substantial additional effort that is beyond the scope of this work. We have further caveated some of our results and added additional details to help further clarify the forcing issue.**

Specific comments:

410-416: You could add to Figure S3 a couple of panels showing the changes in the TA to DIC ratio. We could expect that changes in this variable are closely tight to the $\Omega$, pH, and pCO2 changes (e.g. Wang et al., 2013: https://doi.org/10.4319/lo.2013.58.1.0325).

**RESPONSE: We thank the reviewer for this great suggestion. We have added additional subplots to this figure for the change in the TA/DIC ratio. This further illustrates that the boundary condition change leads to an increase in subsurface TA relative to DIC, which would counteract the subsurface OA signal.**

429-432: This statement is somewhat vague. I agree that the goal of the comparison is not suggesting that the trends are the same for the two forcing products. But you want to support your hypothesis that the enhanced bottom DIC growth in the last period is not an artifact resulting from the forcing change. That is not clear to me from your analysis.

**RESPONSE: We agree that this previous sentence was overly vague and have replaced with the following, expanded description:**

*These amplified bottom water carbonate trends are strongest over the 1998-2022 timeframe, particularly compared to the relatively weak bottom water trends over the 1970-1994 timeframe. This suggests that the strong bottom water trends evolve over the 1998-2022 timeframe, and may be specific to the CFSR forcing. However, the bottom pH trend with CORE forcing increases from -0.0029/decade when calculated from 1970-1994 to -*

*0.011/decade when calculated from 1970-2003 (Fig. S5). Thus, the enhanced bottom water trends over the recent timeframe may be sensitive to the forcing product and the specific timeframe over which the trend is calculated.*

Addressing the second reviewer comment about the greater $\Omega$arag variability you mentioned: "The fact that this is not the case for pH may suggest that temperature is playing a role, since temperature has a relatively greater effect on $\Omega$arag than pH.". I would argue that temperature has a relatively minor impact on $\Omega$arag. You can see that in the Taylor decomposition analysis.

**RESPONSE: This is a good point, we were referring to this in a more general sense (i.e. that temperature has a relatively greater effect on $\Omega_{arag}$ than pH), but the reviewer is correct that in our results, temperature has a relatively minor effect, as demonstrated by the Taylor series decomposition.**

Reviewer 2 Comments

I am satisfied with the additional work and response of the authors. I noticed a few typos or omission in this new version of the manuscript that should be corrected before publication. **RESPONSE: We thank the reviewer for their helpful comments throughout the review process.**

On the new OBC figures the s axis is not explained in the legend.
Some equation numbers were not changed in the text following the addition of the new equations

**RESPONSE: Thank you for catching this, the description of the s-axis has been added to the figure captions and the equation numbers have now been corrected throughout.**

Line 389: TA due to differences (add to)

**RESPONSE: Change made**

Line 405: directly rather than direct
**RESPONSE: Change made**

Lines 652-653: Consistent with (add with)

**RESPONSE: Change made**